# Sperm modulate uterine immune parameters relevant to embryo implantation and reproductive success in mice

John E. Schjenken [1], David J. Sharkey [1], Ella S. Green [1], Hon Yeung Chan [1], Ricky A. Matias[1], Lachlan M. Moldenhauer [1] & Sarah A. Robertson [1]✉

Seminal fluid factors modulate the female immune response at conception to facilitate embryo implantation and reproductive success. Whether sperm affect this response has not been clear. We evaluated global gene expression by microarray in the mouse uterus after mating with intact or vasectomized males. Intact males induced greater changes in gene transcription, prominently affecting pro-inflammatory cytokine and immune regulatory genes, with TLR4 signaling identified as a top-ranked upstream driver. Recruitment of neutrophils and expansion of peripheral regulatory T cells were elevated by seminal fluid of intact males. In vitro, epididymal sperm induced IL6, CXCL2, and CSF3 in uterine epithelial cells of wild-type, but not *Tlr4* null females. Collectively these experiments show that sperm assist in promoting female immune tolerance by eliciting uterine cytokine expression through TLR4-dependent signaling. The findings indicate a biological role for sperm beyond oocyte fertilization, in modulating immune mechanisms involved in female control of reproductive investment.

[1] The Robinson Research Institute and Adelaide Medical School, University of Adelaide, Adelaide, SA, Australia. ✉email: sarah.robertson@adelaide.edu.au

Events at mating exert substantial influence on the likelihood of conception, embryo implantation and pregnancy success[1,2]. Male seminal fluid stimulates the female immune response to provoke a controlled inflammatory response that facilitates embryo implantation[2], and promotes generation of immune tolerance for pregnancy[3,4]. Insufficient immune tolerance is linked with common reproductive disorders in women, including infertility, miscarriage, and pregnancy disorders[3], and unresolved uterine inflammation after mating is associated with poor breeding performance in livestock animals[5]. Pregnancies conceived without seminal fluid contact have compromised outcomes in several mammalian species[2,6]. The female immune changes induced by seminal fluid at conception can in turn affect fetal development, and offspring survival and phenotype[7–10].

Interactions between seminal fluid and the female tract are described in several mammalian and invertebrate species[2,6,11,12]. In mammals, the female response to seminal fluid signals is best characterized in mice and humans[2,6]. We showed previously in mice, that factors in seminal fluid stimulate uterine epithelial cells to trigger an altered gene expression program, with more than 300 genes differentially expressed in the endometrium at 8 h after mating[13]. Cytokine signaling, inflammation and immune response pathways were prominent amongst the genes induced[13]. Ensuing production of pro-inflammatory cytokines and chemokines draws neutrophils, macrophages and dendritic cells from peripheral blood into the underlying endometrial stromal tissue[14,15]. Neutrophils are exuded across the uterine epithelium and into the uterine lumen, where they release neutrophil extracellular traps to eliminate microorganisms and selectively sequester a large proportion of sperm, permitting a smaller subset to retain fertilizing capacity[14,15].

This controlled inflammatory response extends from the cervix and uterus into the higher reproductive tract, to induce oviductal cytokines that support embryo development, and promote progesterone synthesis in the ovary[2]. In the lymph nodes draining the uterus, seminal fluid antigens prime the female adaptive immune response[4,16], eliciting activation and expansion of suppressive regulatory T cells (Treg cells)[17,18]. Treg cells have essential functions at embryo implantation – acting to prevent immune destruction of the foreign embryo, to suppress and resolve inflammation, and to facilitate uterine vascular changes that support optimal placental and fetal development[4,19].

To date, the immune regulatory activity of seminal fluid has been attributed to factors in its plasma fraction, notably transforming growth factor beta (TGFB)[20,21] and E-series prostaglandins[20,22,23]. However, TGFB does not account for the full effect of seminal fluid in female tissues[20,24] and other signaling components likely exist[25,26]. TLR4 signaling is implicated as a key upstream driver of the uterine response, suggesting a signaling role for endogenous ligands of TLR4 in seminal fluid[13].

The extent to which sperm might contribute to evoking the female response to seminal fluid has not been determined. Close physical association between sperm and immune cells and/or epithelial cells lining the female tract occurs in mice[27] and many other species[28]. Sperm attachment to the uterine and oviductal epithelial is a mechanism for sperm sequestration and storage prior to ovulation[28], but whether sperm elicit effects on female immune activation has not been formally considered. We have previously reported that the Treg cell-specific transcription factor *Foxp3*, as well as a critical Treg cell-attracting chemokine *Ccl19*, are more strongly expressed at the time of embryo implantation in the uterus of mice earlier exposed to seminal fluid of intact males, as opposed to vasectomized males[18], implying that stronger immune tolerance may be generated when sperm are present.

Here, we have investigated the specific contribution of sperm in the female response to seminal fluid. Initially, we compared global gene expression in the uterine endometrium of females mated to intact males versus vasectomized males, using Affymetrix microarray. We show that whole seminal fluid containing sperm elicits a pattern of immune response genes and pathways that is distinct to the expression profile elicited by seminal plasma alone, and is associated with elevated neutrophil recruitment into the uterus, and stronger expansion of Treg cells within draining lymph nodes. We then confirm using an in vitro model and *Tlr4* null mutant mice, that sperm specifically induce neutrophil-regulating cytokines IL6, CXCL2, and CSF3 through TLR4 signaling. These data reveal a novel physiological role for sperm at the time of conception and expand understanding of the mechanisms by which seminal fluid interacts with female tissues to generate maternal immune tolerance and reproductive success.

## Results

**Vasectomy alters the female response to seminal fluid after mating.** Previously we showed that seminal fluid induces substantial gene expression changes in the endometrial surface layer of the uterus after mating[13]. We have demonstrated using in vivo and in vitro approaches that soluble signaling factors in seminal plasma regulate several endometrial genes[7,14,20,26]. To evaluate the extent to which sperm contribute to eliciting the female response, global gene expression was examined in endometrial tissue of CBAF1 female mice 8 h following mating with intact males (delivering complete seminal fluid) or vasectomized males (delivering seminal plasma alone). Our previous studies showed strongly upregulated cytokine expression at this time point[7,13]. Unmated virgin estrus females provided a non-mated comparison. Endometrial RNA from 16 females was pooled into 4 independent biological replicates per group, and gene expression profiles were determined by Affymetrix microarray.

Principal component analysis demonstrated a substantial shift in gene expression profile following mating (Fig. 1A), regardless of whether males were intact or vasectomized, in line with previous observations indicating that seminal plasma plays a major role in eliciting the female response[7,14,20,26]. Following mating, a total of 697 genes were upregulated and 434 downregulated (by high stringency criteria of fold-change > 1.5 and FDR $P < 0.05$) in the endometrium of mated females compared to unmated virgin estrus controls (Fig. 1B). Analysis of the most differentially regulated genes (fold-change > 1.5-fold, FDR $P < 0.05$) in both mated groups showed immune system-associated genes to be amongst the most highly regulated, compared to the estrus control group (Supplementary Data 1-2). When differentially regulated genes were assessed by Ingenuity Pathway Analysis, immune signaling pathways were prominent – comprising seven of the top ten signaling pathways identified as activated in either mating group, with a Z-score > 2 (activated) or < −2 (inhibited) ($P < 0.05$) (Supplementary Data 3, Supplementary Table 1).

Endometrial tissue of females mated with intact males exhibited a larger shift in gene expression profile than in females mated with vasectomized males, implicating ejaculate components of testicular and/or epididymal origin in altered gene expression (Fig. 1A). Of the 698 up-regulated genes, 669 were induced following mating with intact males, but only 237 were induced by vasectomized males. Of the 434 down-regulated genes, 420 were suppressed after mating with intact males, compared to 105 suppressed by vasectomized males (Fig. 1B, Supplementary Data 1 and 2).

Direct comparison of endometrial gene expression data from intact- and vasectomized-mated groups showed a total of 110 differentially regulated genes, including 78 induced genes and 32 suppressed genes (Supplementary Data 4). Notably IL6, IL8, and NF-κB signaling were amongst the pathways more strongly activated by intact compared to vasectomized males (Fig. 1C).

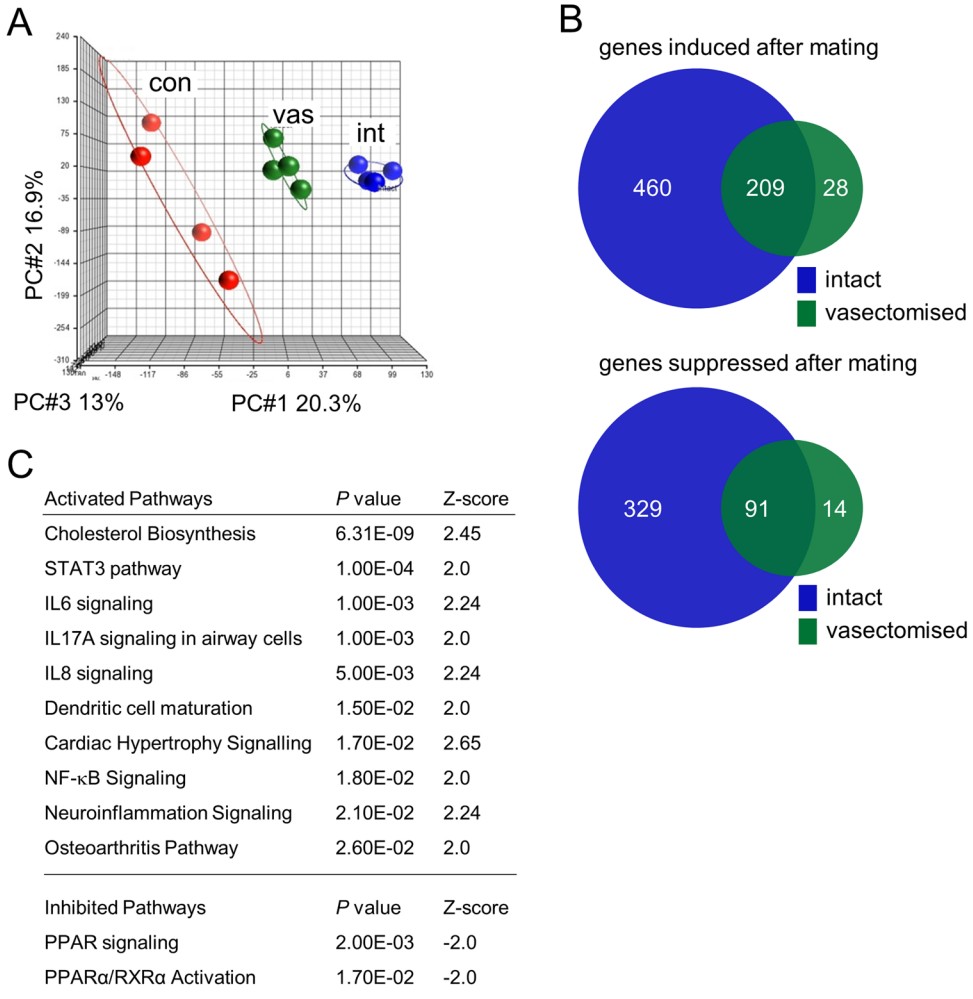

**Fig. 1 Differential gene expression in the uterus after mating with intact males compared to vasectomized males.** Endometrial tissue was collected from females in estrus (unmated control, con), or 8 h after mating with either intact (int) or vasectomized (vas) males. RNA was extracted and gene expression measured by Affymetrix Mouse Gene 1.0 ST arrays ($n = 4$ biological replicates, pooled from a total of 16 tissue samples per group). Microarray data was analyzed by Partek Genomics Suite and Ingenuity Pathway Analysis. Principal component analysis of microarray data according to group (**A**). Venn diagrams show the number of genes significantly induced or suppressed (>1.5-fold change and FDR < 0.05) following mating (**B**). Canonical signaling pathways that are activated (Z-score > 2) or inhibited (Z-score < −2) ($P$ < 5.0E-02) in the endometrium of females mated with intact males compared to vasectomized males (**C**).

**Endometrial cytokine genes are differentially regulated by vasectomized versus intact males.** Several pro-inflammatory cytokines and chemokines were amongst the differentially regulated genes (18 upregulated, 1 down-regulated, fold-change >1.4-fold) in endometrium of intact-mated vs. vasectomized-mated females (Table 1). To validate the microarray data, 13 of these 19 genes were quantified by qPCR. Consistent with the microarray data, *Il6*, *Cxcl2*, *Csf3,* and *Cxcl1* (Fig. 2A-D), as well as *Ccl3*, *Ccl21a*, *Cxcl5*, *Cxcl7*, *Cxcl10*, *Il1b*, *Lif*, and *Tnf* (Supplementary Fig. 1) were all regulated differentially by mating with intact compared to vasectomized males. Amongst these, *Il6* (Fig. 2A), *Cxcl7*, *Il1b*, and *Lif* (Supplementary Fig. 1) were induced by mating with intact, but not vasectomized males. *Cxcl2*, *Csf3*, *Cxcl1* (Fig. 2B-D), *Ccl3*, *Cxcl5*, *Cxcl10*, and *Tnf* (Supplementary Fig. 1) were induced by mating in both groups, but to a higher degree in the intact group. Two genes, *Csf1* and *Il1a* (Supplementary Fig. 1), were induced to a similar degree in intact- and vasectomized-mated females. Both microarray and qPCR data showed *Ccl21a* was inhibited by mating with intact males, but unchanged after mating with vasectomized males (Supplementary Fig. 1, Supplementary Data 1 and 2).

Additional genes not impacted by male status in the microarray data, but previously shown to be regulated by seminal fluid[13,14], were also assessed. *Ccl2*, *Ccl20*, *Ccl22*, and *Csf2* were all induced to a greater extent in the intact-mated compared to vasectomized-mated group (Supplementary Fig. 1).

Analysis by t-distributed stochastic neighbor embedding (tSNE) (Supplementary Fig. 1) showed the qPCR data clustered according to mating status, with distinct unmated, intact-mated and vasectomized-mated groups. The intact-mated group was shifted more distinctly and to a greater extent from the unmated group, than the vasectomized-mated group. Together these data indicate that mating with intact males elicits a greater change in endometrial cytokine gene expression and immune response gene pathways than does mating with vasectomized males, implying a role for sperm and/or epididymal factors.

**Endometrial cytokine proteins are differentially regulated by vasectomized versus intact males.** To investigate the impact of seminal fluid composition on endometrial cytokines and chemokines, proteins were quantified by multiplex bead analysis in fluid

**Table 1 Cytokine and chemokine genes differentially expressed (>1.4-fold fold change) in endometrial tissue of intact (int) compared to vasectomized (vas) mated females, as identified by microarray.**

| | | int vs. vas | | int vs. con | | vas vs. con | |
|---|---|---|---|---|---|---|---|
| Gene Symbol | RefSeq | FC | FDR | FC | FDR | FC | FDR |
| *Ccl3 | NM_011337 | 2.43 | 2.70E-01 | 3.11 | 7.67E-02 | 1.28 | 7.91E-01 |
| *Ccl21a | NM_011124 | −3.49 | 2.74E-02 | −3.80 | 2.23E-03 | −1.09 | 8.70E-01 |
| Ccl28 | NM_020279 | 1.43 | 2.26E-01 | 1.92 | 1.37E-02 | 1.34 | 2.85E-01 |
| *Csf1 | NM_007778 | 1.66 | 1.67E-01 | 2.74 | 4.59E-03 | 1.65 | 1.39E-01 |
| *Csf3 | NM_009971 | 2.08 | 3.26E-02 | 6.20 | 8.62E-05 | 2.98 | 3.70E-03 |
| *Cxcl1 | NM_008176 | 1.98 | 3.26E-02 | 3.38 | 3.14E-04 | 1.71 | 3.45E-02 |
| *Cxcl2 | NM_009140 | 2.77 | 1.32E-01 | 23.58 | 3.39E-04 | 8.50 | 8.19E-03 |
| Cxcl3 | NM_203320 | 1.56 | 7.10E-02 | 1.97 | 3.20E-03 | 1.26 | 2.65E-01 |
| *Cxcl5 | NM_009141 | 4.20 | 4.24E-02 | 23.59 | 2.23E-04 | 5.62 | 1.26E-02 |
| *Cxcl7 | NM_023785 | 2.06 | 2.40E-01 | 2.69 | 4.87E-02 | 1.30 | 6.88E-01 |
| *Cxcl10 | NM_021274 | 1.80 | 5.17E-01 | 6.36 | 1.54E-02 | 3.53 | 1.18E-01 |
| Cxcl16 | NM_023158 | 1.57 | 1.00E-01 | 2.51 | 1.46E-03 | 1.60 | 6.46E-02 |
| *Il1b | NM_008361 | 3.81 | 4.50E-02 | 5.54 | 3.03E-03 | 1.46 | 4.96E-01 |
| *Il6 | NM_031168 | 1.46 | 4.02E-02 | 1.70 | 1.57E-03 | 1.16 | 2.97E-01 |
| Il17c | NM_145834 | 3.47 | 7.35E-02 | 12.53 | 8.18E-04 | 3.61 | 4.34E-02 |
| Il23a | NM_031252 | 1.66 | 6.41E-02 | 1.81 | 8.83E-03 | 1.09 | 7.52E-01 |
| *Lif | NM_008501 | 2.60 | 5.08E-03 | 1.44 | 2.61E-01 | 1.80 | 1.08E-01 |
| Ptgs2 | NM_011198 | 1.91 | 3.72E-01 | 7.03 | 5.06E-03 | 3.68 | 6.15E-02 |
| *Tnf | NM_013693 | 2.31 | 1.18E-01 | 5.31 | 2.23E-03 | 2.30 | 9.14E-02 |

Comparisons are; genes differentially regulated between females mated with intact and vasectomized males (int vs.vas); between females mated with intact males and virgin estrus control (con) females (int vs. con), and between females mated with vasectomized males and control females (vas vs. con). FC = fold-change; FDR = false discovery rate $P$ value. *Confirmatory qPCR analysis was undertaken on several genes (Fig. 1 and Supplementary Fig. S1).

from the uterine lumen collected 8 h after mating with intact or vasectomized males, or from unmated estrus controls. IL6, CXCL1 (Fig. 2E, H), CCL2, CSF1, and CXCL10 (Supplementary Fig. 2) were increased after mating, and were more highly induced by intact compared to vasectomized males. CSF3 was significantly induced in the intact but not vasectomized group (Fig. 2G). In contrast, CXCL2 (Fig. 2F), CSF2, CXCL5, CXCL7, IL1B, LIF, and TNF (Supplementary Fig. 2) were induced to equivalent levels in both mating groups. Similar data were seen in solubilized endometrial tissue, with elevated abundance of CSF3 (Fig. 2K), CCL2, and CXCL10 (Supplementary Fig. 3) at 8 h after mating with intact males, compared to vasectomized males, and induction of CXCL1, CXCL2 (Fig. 2J, L), and LIF (Supplementary Fig. 3) after mating with intact, but not vasectomized males, but similar abundance of IL6 in both mated groups (Fig. 2I). tSNE analysis of the luminal fluid data showed clusters in the uterine cytokine profile associated with mating group (Supplementary Fig. 2). These data provide evidence of differential induction of endometrial cytokines attributable to sperm and/or epididymal factors.

**Vasectomy modulates seminal fluid-induced uterine neutrophil influx after mating.** Amongst the endometrial cytokines differentially induced by intact compared to vasectomized males, several are associated with neutrophil recruitment and function (Supplementary Fig. 4). To assess neutrophils, uterine tissue was collected from females mated with vasectomized or intact males and neutrophils were detected by immunostaining for the neutrophil marker Ly6G[29]. Neutrophil influx into the uterus was strongly evident at 8 h after mating with intact males (Fig. 3A), when Ly6G+ cells with a granulocytic appearance accumulated in both the myometrium and particularly the endometrium of most females. In contrast, uterine neutrophils were substantially fewer after mating with vasectomized males, and were notably deficient in the endometrial stroma immediately subjacent to the luminal and glandular epithelium (Fig. 3B, D). Ly6G+ neutrophils were 12-fold more abundant in the endometrium of intact-mated females than vasectomized-mated females (Fig. 3B, D) ($P < 0.05$).

Large numbers of Ly6G+ neutrophils and extensive complexes of sperm engulfed in Ly6G+ neutrophil extracellular traps (NETs) were also abundant in the luminal cavity as detected by staining with NET markers DAPI and Ly6G[30] (Fig. 3E, F). Neither neutrophils nor NETs were present in the luminal fluid of estrus females or females mated with vasectomized males.

**Vasectomy modulates seminal fluid-induced Treg cell induction after mating.** A key consequence of mating is activation and expansion of Treg cells in lymph nodes draining the uterus to facilitate tolerance of embryo implantation[4]. To investigate the effect of seminal fluid composition on Treg generation, CD4+ T cells, CD4+FOXP3+ Treg cells, CD4+FOXP3+NRP1+ thymic Treg (tTreg) cells, and CD4+FOXP3+NRP1− peripheral Treg (pTreg) cells were evaluated by flow cytometry (Supplementary Fig. 5) in uterine draining lymph nodes collected following mating with vasectomized or intact males, on day 3.5 post-coitum (pc) when peak T cell proliferation is observed[16], and from estrus females. A comparable rise in the number of CD4+ T cells occurred after mating with vasectomized (1.7-fold, $P = 0.054$) and intact (2.0-fold, $P < 0.01$) males (Fig. 4A). While both mated groups showed an increase in pTreg, tTreg and total Treg cell numbers, intact-mated females had a 50% higher expansion of CD4+FOXP3+NRP1−/lo pTreg cells compared to females mated with vasectomized males (Fig. 4B, $P < 0.05$), while comparable numbers of tTreg cells and total Treg cells (pTreg + tTreg cells) were induced (Fig. 4C, D). Analysis of activation marker CD25, proliferation marker Ki67, and suppressive competence marker CTLA4 in Treg cells, did not reveal substantial differences in the Treg phenotype attributable to male status, although CTLA4 was consistently upregulated relative to unmated control only by intact males, not vasectomized males (Supplementary Fig. 6).

**Sperm elicits cytokine synthesis in mouse uterine epithelial cells.** These data imply that sperm or soluble seminal fluid factors originating in the testes or epididymis contribute to eliciting the uterine immune response to seminal fluid. To specifically

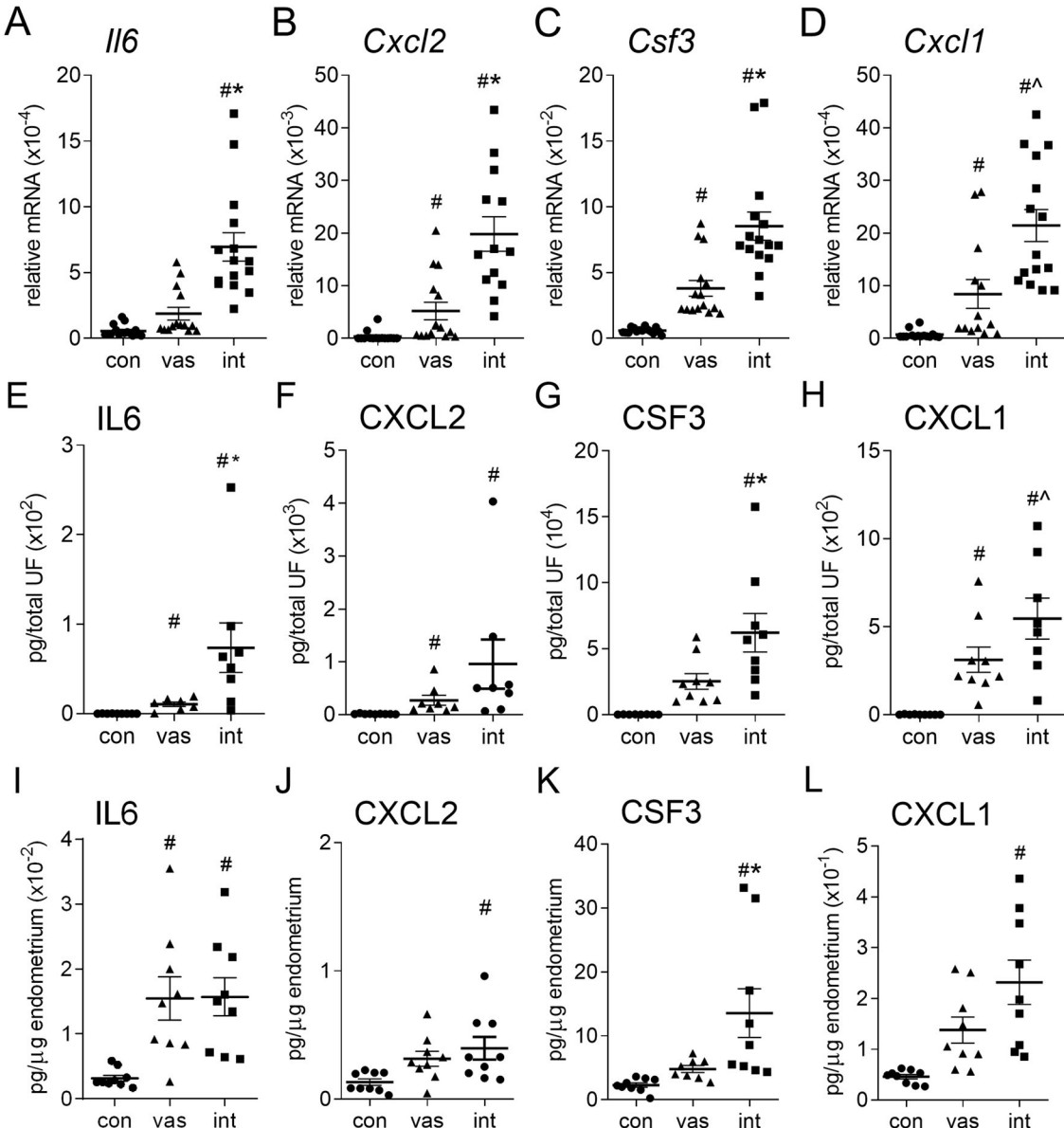

**Fig. 2 Cytokine synthesis in the uterus after mating with intact males compared to vasectomized males.** The endometrial layer of the uterus and uterine luminal fluid was collected from females in estrus (unmated control, con), or 8 h after mating with either intact (int) or vasectomized (vas) males. RNA was extracted, transcribed into cDNA, and expression of *Il6* (**A**), *Cxcl2* (**B**), *Csf3* (**C**), and *Cxcl1* (**D**) were quantified by qPCR ($n = 13–16$/group). IL6 (**E**, **I**), CXCL2 (**F**, **J**), CSF3 (**G**, **K**), and CXCL1 (**H**, **L**) protein were quantified by multiplex microbead assay in uterine luminal fluid (**E–H**) and endometrial tissue (**I–L**) ($n = 8–9$/group). Symbols depict data from individual mice and mean ± SEM values are shown. Data were analyzed by one-way ANOVA with post-hoc Sidak t-test, or non-parametric Kruskal Wallis test with post-hoc Dunn's multiple comparisons test. $^{\#}P < 0.05$ compared to unmated estrus control group, $^{*}P < 0.05$ compared to vasectomized-mated group.

investigate the relative contribution of sperm, an in vitro model was utilized. Mouse uterine epithelial cells from estrous mice were cultured with epididymal sperm, seminal vesicle fluid as a comparison, or a combination of both, using an established methodology and time course[26].

Incubation with sperm for 16 h elicited a consistent increase in IL6 (maximum 1.8-fold, Fig. 5A), CXCL2 (1.5-fold, Fig. 5D), and CSF3 (1.6-fold, Fig. 5G), but negligible effect on CXCL1 (Fig. 5J). Other cytokines induced in vivo, CSF2, TNF, LIF, and CCL2, were all inhibited by 16 h culture with sperm, in a dose-dependent manner (Supplementary Fig. 7). In contrast, 16 h culture with seminal vesicle fluid elicited increases only in IL6 (1.3-fold, Fig. 5B), CXCL1 (2.7-fold, Fig. 5K), and LIF (Supplementary Fig. 7). By 40 h after seminal vesicle fluid contact, all cytokines including CXCL2

(Fig. 5E), CSF3 (Fig. 5H), and CXCL1 (Fig. 5K), as well as CSF2, TNF, LIF and CCL2 (Supplementary Fig. 7), were induced, other than IL6, which was suppressed after longer culture (Fig. 5B).

When sperm and seminal plasma were added together to uterine epithelial cells, an additive interaction between sperm and seminal plasma was seen for IL6 (Fig. 5C) and CSF3 (Fig. 5I) but not CXCL1 (Fig. 5L). Notably CXCL2 and CCL2 were substantially further elevated, by 21-fold (Fig. 5F) and 29-fold (Supplementary Fig. 7), indicating a synergistic interaction when both components were present.

**Identification of TLR4 as a key mediator of the female response to sperm exposure.** Next, we used Ingenuity Pathway Analysis to

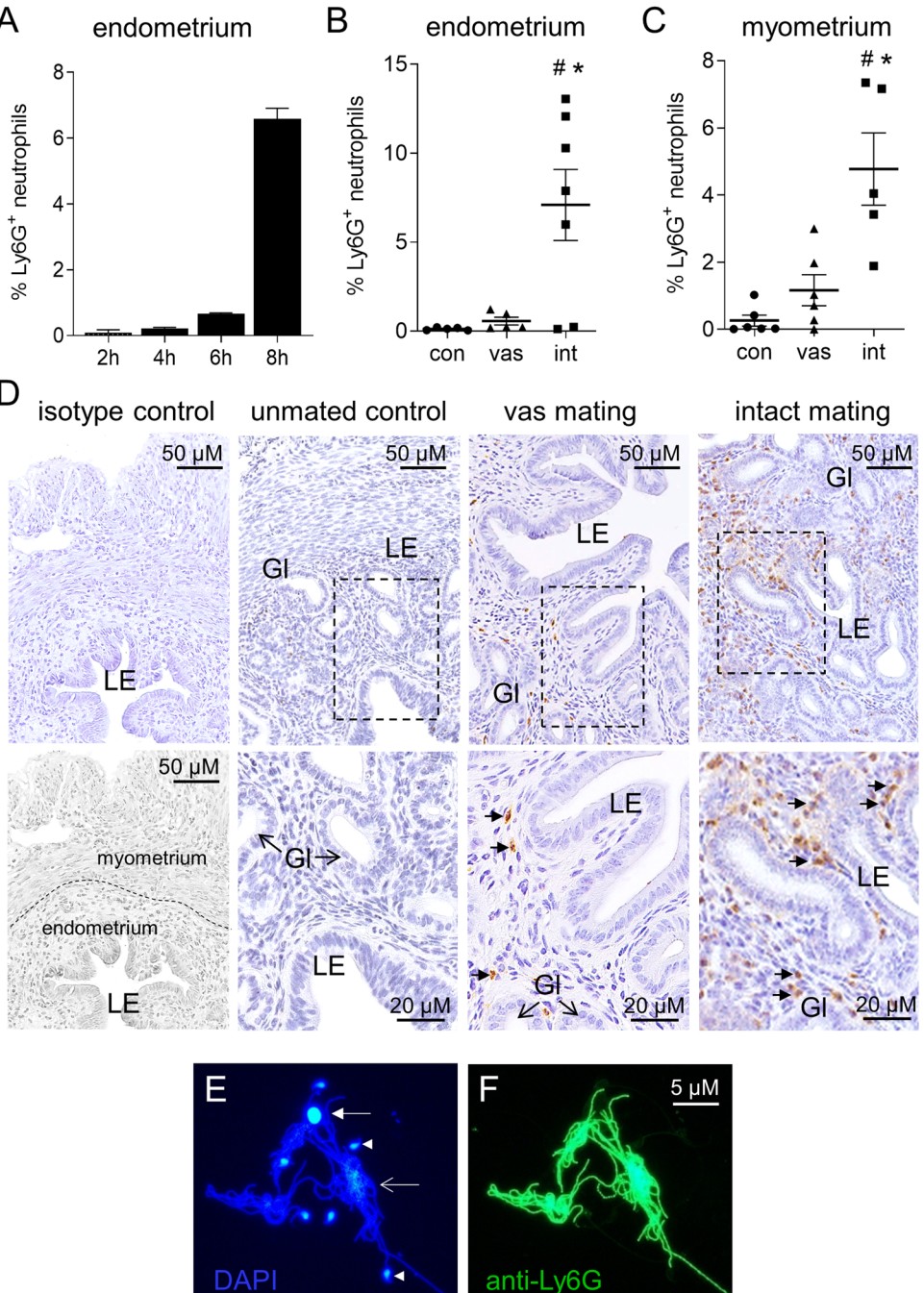

**Fig. 3 Neutrophil recruitment after mating with intact males compared to vasectomized males.** To evaluate neutrophils, uterine tissue was collected from females in estrus (unmated control, con), or after mating with either intact (int) or vasectomized (vas) males, and Ly6G+ neutrophils were detected by immunohistology in the uterine endometrium, myometrium, and luminal fluid. Ly6G+ neutrophils in the endometrium at 2, 4, 6 and 8 h following mating with intact males (**A**), and in the endometrium (**B**) and myometrium (**C**) at 8 h after mating with intact or vasectomized males. Symbols depict data from individual mice ($n = 5$–7/group) and mean ± SEM values are shown. Data were analyzed by one-way ANOVA with post-hoc Sidak t-test. #$P < 0.05$ compared to unmated estrus control group, *$P < 0.05$ compared to vasectomized-mated group. Representative images of Ly6G+ neutrophils (small arrows) in the endometrium of unmated control, vasectomized-mated (vas) and intact-mated (int) females, at high and low power (**D**). Isotype-matched negative control, with myometrium, endometrium, and luminal epithelium compartments, is also shown. LE = luminal epithelium; Gl = epithelial glands. Neutrophils and neutrophil extracellular traps were abundant in uterine luminal fluid from females mated with intact males by staining with DAPI (**E**) and Ly6G (**F**), but were not detected in luminal fluid of unmated control or vasectomized-mated females. Closed arrow = intact nuclei of a neutrophil; open arrow = apoptotic neutrophil; arrow head = sperm head.

identify the upstream signaling pathways involved in transmitting the differential effects of mating with intact versus vasectomized males. A total of 14 activated and 3 inhibited upstream regulators were identified, on the basis of a Z-score > 2 or Z-score < −2

(Supplementary Table 2). These upstream regulators were all more highly activated or inhibited following mating with intact males, compared to vasectomized males (Fig. 6A, Supplementary Data 5 and 6). Amongst the activated upstream regulators were

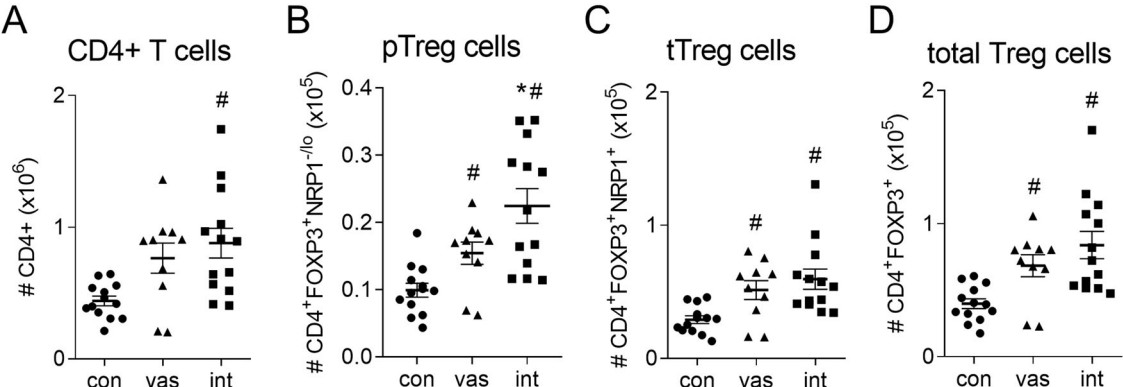

**Fig. 4 Treg cell generation after mating with intact males compared with vasectomized males.** To evaluate Treg cells, uterus-draining para-aortic lymph nodes (PALN) were collected from female mice in estrus (unmated control, con), or on day 3.5 pc after mating with either intact (int) or vasectomized (vas) males. CD4$^+$ T cells were analyzed by flow cytometry for CD4, FOXP3 and NRP1 expression, and the numbers of CD4$^+$ T cells (**A**), CD4$^+$FOXP3$^+$ NRP1$^{-/lo}$ peripheral Treg cells (**B**), CD4$^+$FOXP3$^+$NRP1$^+$ thymic Treg cells (**C**), and total CD4$^+$FOXP3$^+$ Treg cells (**D**) were quantified. Symbols depict data from individual mice ($n = 10$–13/group) and mean ± SEM values are shown. Data were analyzed by one-way ANOVA with post-hoc Sidak t-test, or non-parametric Kruskal Wallis test with post-hoc Dunn's multiple comparisons test. $^{\#}P < 0.05$ compared to unmated estrus control group, $^*P < 0.05$ compared to vasectomized-mated group.

factors previously associated with seminal plasma signaling, including TLR4/MYD88[13], CD38[25], and IFNG[26]. Of these molecules, both MYD88 and TLR4 were in the top 3 predicted activated upstream regulators (Fig. 6A) and were also identified as central network regulators involved in neutrophil recruitment (Supplementary Fig. 4), implicating TLR4 ligands carried by sperm as key mediators of seminal fluid signaling activity.

**Impact of TLR4 null mutation on the uterine cytokine and neutrophil response after mating.** To determine whether TLR4 is critical for uterine responsiveness to seminal fluid factors, uterine epithelial cells were harvested from $Tlr4^{-/-}$ and $Tlr4^{+/+}$ females. Addition of sperm to $Tlr4^{+/+}$ uterine epithelial cells elicited increased IL6 (Fig. 6B), CXCL2 (Fig. 6C), and CSF3 (Fig. 6D), but this was not observed following culture of sperm with cells from $Tlr4^{-/-}$ females. Again, addition of sperm in vitro did not induce CXCL1 (Fig. 6E), CCL2, CSF2, LIF, or TNF (Supplementary Fig. 8).

To determine whether TLR4 is critical for sperm-induced neutrophil recruitment into the uterine endometrium, uteri recovered from $Tlr4^{-/-}$ and $Tlr4^{+/+}$ females mated with intact wild-type males and Ly6G$^+$ neutrophils were detected by immunostaining[29]. Ly6G$^+$ neutrophil influx into the uterine myometrium was clearly evident at 8 h after mating in both $Tlr4^{-/-}$ and $Tlr4^{+/+}$ females (Fig. 7A, C), but neutrophils were 75% and 78% reduced in endometrial tissue, particularly in the sub-epithelial stroma, of $Tlr4^{-/-}$ females (Fig. 7B, C) ($P < 0.05$), This indicates that TLR4 is essential for sperm-mediated attraction of neutrophils into the endometrium and luminal cavity.

## Discussion
Evidence is building that attributes of seminal fluid other than sperm fertilizing capability influence reproductive outcomes in mammals[1,10]. This is due to seminal fluid signaling elements that modulate the female immune response to affect embryo development and implantation[7,9]. The current study demonstrates that ejaculated sperm play an active role in provoking the female immune response, by interacting with uterine epithelial cells to induce cytokines that assist in eliciting neutrophil recruitment and T cell activation. We found the female response was blunted after mating with vasectomized males, with impaired cytokine induction, neutrophil accumulation, and Treg cell generation. A direct action of sperm in signaling to female tissues was

confirmed by in vitro experiments, where cytokines IL6, CXCL2, and CSF3 were produced by uterine epithelial cells incubated with epididymal sperm, via a TLR4-dependent mechanism. Together these results indicate that sperm interacting with uterine epithelial cells help initiate the female immune response in early pregnancy.

Close physical association between sperm and epithelial cells has been thought to sustain a reservoir of sperm capable of fertilizing oocytes, or to assist in the removal of superfluous sperm[28]. The question of whether sperm-epithelial interactions impact the immune response or other aspects of female reproductive physiology has rarely been considered. In bovine, culture of sperm with endometrial fragments was shown to induce expression of cytokines in endometrial epithelial cells including TNF, IL1B, and IL8[31] and this was suppressed by co-incubation with neutralizing antibodies to TLR2 and TLR4, indicating a possible role for TLR4[32]. In pigs, insemination with sperm-rich fractions of the ejaculate caused greater changes in endometrial gene expression than seminal plasma alone, attenuating immune response pathways amongst others[33], however in utero transfer of washed sperm did not induce endometrial cytokines in vivo[34], and others reported a reduced uterine cytokine response when sperm were added to seminal plasma[35]. The potential for sperm-epithelial interactions to induce changes that facilitate maternal immune tolerance, or otherwise promote endometrial receptivity, has not been investigated.

Our previous study reported that seminal fluid induces an array of uterine inflammatory cytokines and chemokines at the gene transcription level[13], but we did not define the individual contribution of sperm versus seminal plasma. The current finding of strong uterine induction of immune response genes after mating with vasectomized males is consistent with extensive evidence that seminal plasma elicits uterine cytokine expression[7,14,20,26], and vasectomized males can stimulate female Treg cell generation conferring functional tolerance[4,18]. The finding that sperm also contribute to inducing uterine cytokine genes is novel.

Surgical ligation of the vas deferens results in ejaculation of seminal fluid that differs from an intact ejaculate by the absence of sperm, and is also missing secretions from the testes, epididymis and vas deferens. Whether epididymal secretions interact with female tissues in vivo is not clear[36], but is likely given that extracellular vesicles in seminal fluid can modulate gene

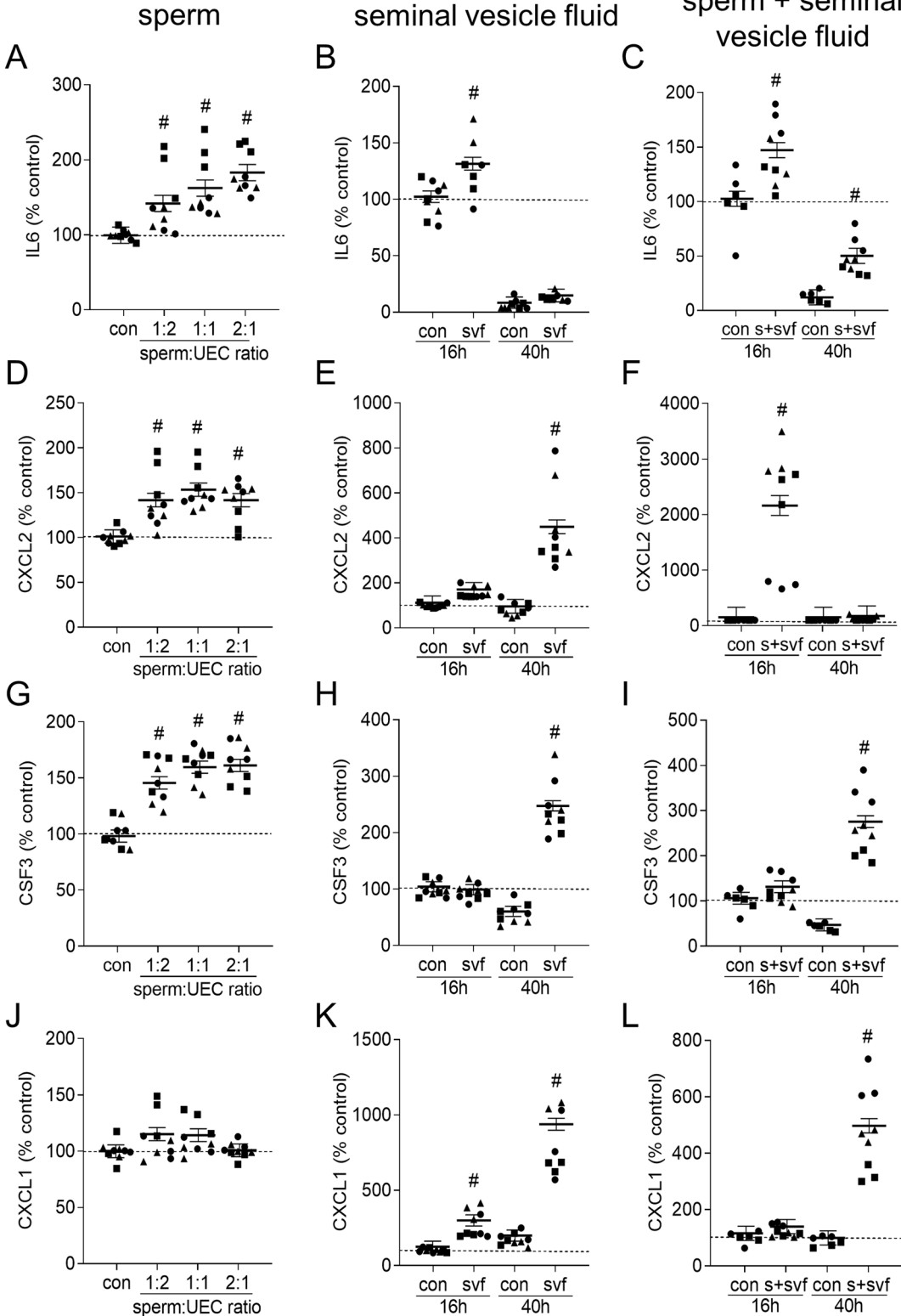

**Fig. 5 Sperm induction of cytokine release from uterine epithelial cells in vitro.** Uterine epithelial cells (UEC) were isolated from estrus CBAF1 mice and incubated with control media alone (con), and sperm (1:2, 1:1, 2:1 sperm:UEC ratio) (**A**, **D**, **G**, **J**), seminal vesicle fluid (svf; 5% total culture volume) (**B**, **E**, **H**, **K**), or a combination of 2:1 sperm:UEC + 5% svf (**C**, **F**, **I**, **L**), for 16 h. Culture media was replaced and collected following an additional 24 h culture period, then IL6 (**A–C**), CXCL2 (**D–F**), CSF3 (**G–I**), and CXCL1 (**J–L**) were quantified by multiplex microbead assay in supernatants recovered at 16 h (**A–L**) and 40 h (**B**, **C**, **E**, **F**, **H**, **I**, **K**, **L**). Mean baseline cytokine concentration in control supernatants was 1.3 ng/$10^5$ cells/24 h (IL6); 8.5 ng/$10^5$ cells/24 h (CXCL2), 14 ng/$10^5$ cells/24 h (CSF3), and 9 ng/$10^5$ cells/24 h (CXCL1). Symbols depict data from 3 experiments (with 3 replicates/experiment), and estimated marginal mean ± SEM values are shown. Data was analyzed by mixed model ANOVA, with experiment as covariate. #$P < 0.05$ compared to medium alone control group.

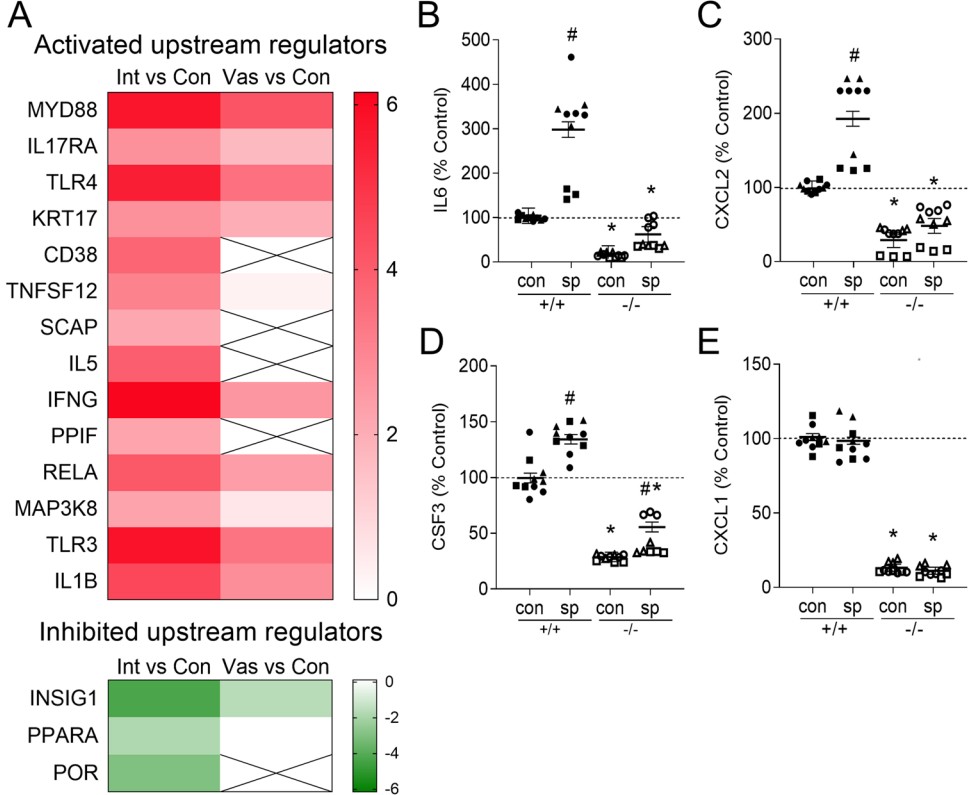

**Fig. 6 Sperm utilize TLR4 to regulate uterine gene expression and induce cytokine release.** Ingenuity Pathway Analysis was used to predict significantly ($P < 0.05$) activated (Z-score > 2) or inhibited (Z-score < −2) upstream regulators of the differentially expressed endometrial genes after mating with intact versus vasectomized males, in the microarray analysis. A heat map of upstream regulators based on Z-score to show differences induced by intact and vasectomized males, compared to estrus control, with intensity of colouration indicating degree of activation (z-score > 2 with color ranging from red (Z-score = 6) to white (Z-score = 0) or inhibition (z-score < −2 with color ranging from green (Z-score = −6) to white (Z-score = 0)). Crossed boxes signify factors not predicted to be regulators in that data set. (**A**). Uterine epithelial cells (UEC) were isolated from estrus $Tlr4^{+/+}$ or $Tlr4^{-/-}$ mice and incubated with control media alone (con), or sperm (2:1 sperm:UEC ratio) (**B**–**E**), for 16 h. Supernatants were collected then IL6 (**B**), CXCL2 (**C**), CSF3 (**D**), and CXCL1 (**E**) were quantified by multiplex microbead assay. See legend to Fig. 4 for mean baseline cytokine concentration in control supernatants. Symbols depict data from 3 experiments (with 3 replicates/experiment), and estimated marginal mean ± SEM values are shown. Data was analyzed by mixed model ANOVA, with experiment as covariate. #$P < 0.05$ compared to medium alone control group, *$P < 0.05$ compared to $Tlr4^{+/+}$ cells with same treatment.

expression in female reproductive tract cells in vitro[37,38]. It is also possible that surgical intervention to perform vasectomy causes subtle changes to male accessory gland secretions. However, our in vitro experiment clearly demonstrates that sperm can bind and transmit signals to uterine epithelial cells, to elicit increased cytokine synthesis. The sperm recovery protocol utilized for the in vitro experiment resulted in extensive (~2000-fold) dilution of contaminating epididymal secretions, so reasonably they would not account for cytokine induction. Collectively, therefore, these data provide compelling evidence of a sperm-mediated effect, and imply that the constrained response to vasectomized males is at least partly attributable to interactions between sperm and the uterine epithelium.

Previous studies show that both human and mouse sperm carry signaling moieties with the potential to directly engage TLR4[28]. The identity of these agents remains to be determined, but endogenous danger-associated molecular patterns (DAMPs) that bind and signal through TLR4 are likely candidates. Various DAMPs are present on sperm including hyaluronidase[39], β-defensins[40], and sialic acid and other glycans[41]. Another candidate is the a2 isoform of V-ATPase, which regulates uterine *Ccl2*, *Il1b*, *Lif*, and *Tnf* expression after release from capacitated sperm[42]. The model TLR4 ligand lipopolysaccharide (LPS) can induce cytokine release from uterine epithelial cells in vitro, but there is insufficient LPS in seminal fluid to be responsible[13].

Furthermore, *Tlr4* null mutant female mice exhibit reduced fertility and poor pregnancy outcomes[43]. Whether this is due to inability to respond to sperm signals cannot readily be determined, since seminal plasma also interacts with uterine epithelial cells through TLR4[13].

Sperm may acquire signaling moieties via epididymosomes which accumulate on the sperm head during epididymal transit[36,44] delivering cargo including immune-modulating microRNAs and proteins[44–46]. Seminal plasma components absorbed onto sperm after ejaculation could also mediate sperm interactions with the female tract. This raises a limitation of the in vitro experiments, which used epididymal sperm that could have minor differences in surface properties compared to ejaculated sperm. Seminal fluid extracellular vesicles exhibit immune-regulatory functions[45] and in vitro studies in pigs and humans suggest vesicles are capable of eliciting transcriptional modulation in female tissues[37,38], including upregulation of *CCL20* in uterine epithelial cells[37]. Differential exposure to extracellular vesicles in ejaculated versus epididymal sperm could account for why some cytokines elicited by intact males in vivo, such as CXCL1, CSF2, TNF, LIF, and CCL2, were not induced by sperm in vitro. Sperm interaction with prostasomes or epididymosomes could also explain why CXCL2 and CCL2 were maximally elevated when both sperm and seminal plasm were added to epithelial cells, indicating a synergistic effect (Table 2). We did not find uterine

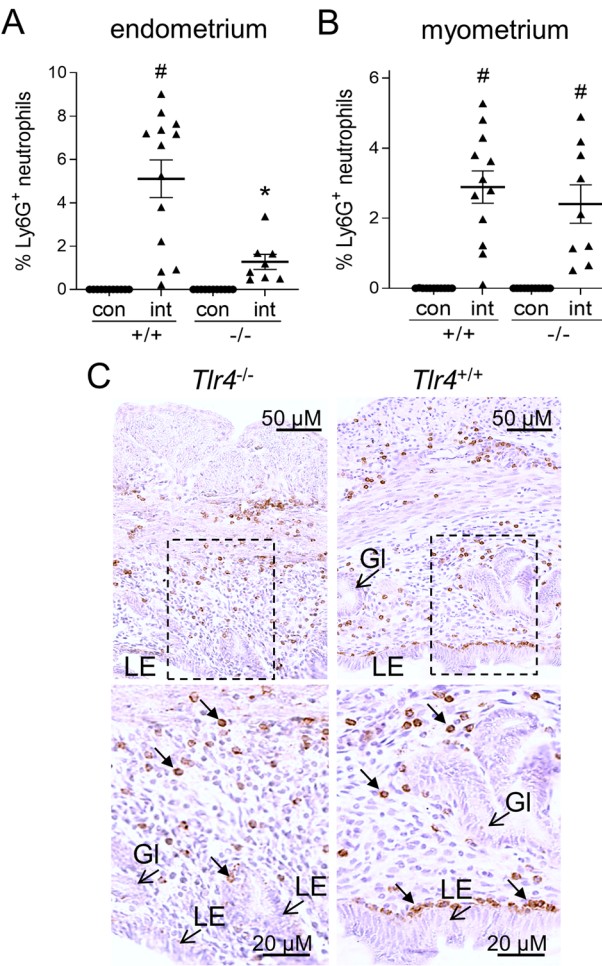

**Fig. 7 TLR4 deficiency impairs sperm-induced neutrophil recruitment into the uterine endometrium.** To evaluate neutrophils, uterine tissue was collected from $Tlr4^{-/-}$ female ($-/-$) and $Tlr4^{+/+}$ ($+/+$) females in estrus (con), or 8 h after mating with intact males (int), and Ly6G+ neutrophils were detected by immunohistology. Ly6G+ neutrophils were quantified in endometrium (% positivity) (**A**) and myometrium (% positivity) (**B**). Symbols depict data from individual mice ($n = 9$-13/group) and mean ± SEM values are shown. Data were analyzed by one-way ANOVA with post-hoc Sidak t-test, #$P < 0.05$ compared to unmated estrus control group, *$P < 0.05$ versus $Tlr4^{+/+}$ females. Representative images of Ly6G+ neutrophils (small arrows) in endometrium of mated $Tlr4^{-/-}$ females ($-/-$) and $Tlr4^{+/+}$ ($+/+$) females, at high and low power (**C**). LE luminal epithelium, GI epithelial glands.

cytokines where sperm and seminal plasma exerted opposing effects, as was suggested by a previous study in pigs[35].

Many studies report extensive interactions between sperm and neutrophils in the uterine lumen after mating in mice[14] and livestock animal species[5], but whether neutrophil influx specifically requires sperm has not been clear. Artificial insemination experiments using washed sperm in pigs and horses imply that sperm directly attract neutrophils into the uterine lumen[47,48], and sperm appear to cause the post-coital influx of cervical neutrophils in women[49]. In vitro experiments show neutrophil chemoattractants intrinsic to sperm and seminal plasma are partly responsible[48,50,51]. However, the current study implies that neutrophil chemokines released from the uterus in response to sperm are essential to eliciting the full neutrophil response. IL6, CXCL2, and CSF3, each confirmed to be induced by sperm in vitro, are all potent regulators of neutrophil recruitment and functional

activation[52,53]. This depends on TLR4 expression, since females genetically deficient in TLR4 showed substantially reduced neutrophil accumulation in the endometrial stroma and luminal epithelium. Consistent with this, we have shown that uterine induction after mating of IL6, CXCL2, and CSF3 fails to occur in TLR4-deficient females[13]. Together, these data strongly indicate that sperm induce uterine chemotactic factors to amplify the neutrophil response and position neutrophils at the luminal epithelial surface for trans-epithelial migration into the luminal cavity. Amplification through female tract cytokine gene induction fits with the dynamics of uterine neutrophil influx, which begins within the first hour of mating, but doesn't peak until several hours later[5,14,15].

Once in the luminal cavity, neutrophils selectively eliminate subsets of sperm by release of NETs[51] and trogocytosis[54], although the basis of selection remains unclear. Although this is can be viewed as a 'hostile' female response to the male gametes[28], neutrophil-mediated selection of the fittest or most compatible sperm for fertilization may ultimately increase the chances of a given male siring healthy offspring[55]. A substantial population of neutrophils recruited at mating are retained in the endometrium and accumulate in implantation sites[56,57], where they potentially regulate various aspects of embryo implantation through tissue remodeling and immune regulation, particularly by linking the innate and adaptive immune compartments[52,58].

As well as targeting neutrophils, IL6, and CSF3 regulate other aspects of uterine receptivity[59,60], and mice deficient in either cytokine have reduced fertility[59,61]. Dysregulated endometrial expression of IL6[62] or CSF3[63] is linked with unexplained infertility and recurrent miscarriage in women. The role of CXCL2 in the uterus is less well defined, but it is temporally modulated over the reproductive cycle[64]. The effects of sperm contact on cytokines associated with implantation may persist for several days—$Ccl2$, $Lif$, and $Ccl19$ remain elevated in uterine tissue four days after mating with intact males, compared to vasectomized males[18,42].

The reduction in pTreg cells after mating with vasectomized males compared to intact males indicates a role for sperm in generating Treg cells. pTreg cells are important mediators of maternal immune tolerance and are rate-limiting in embryo implantation and fetal survival[65], especially in second and subsequent pregnancies[66]. An effect of sperm is consistent with our previous report of maximal induction of the Treg transcript $Foxp3$ in the uterus of females exposed to whole seminal fluid[18]. pTregs are induced from naive T helper cells responding to paternal allo-antigens, and are thought to suppress the generation of anti-fetal as well as anti-sperm immunity, thereby promoting survival of sperm and concepti after future matings. Fewer pTregs after mating with vasectomized males could be a consequence of lack of sperm antigen[67], or reduced CSF3, since CSF3 acts to induce tolerogenic dendritic cells[68] required to present antigen to pTreg cells[69]. Sperm-induced neutrophils may help to drive pTreg proliferation, since neutrophil-dependent Treg cells are involved in placental development[70].

There is clear evidence that contact with conceiving partner's seminal fluid improves the chances of conception and pregnancy in the clinical IVF setting[71]. The current data raise the question of whether human sperm have similar cytokine-inducing activity, and the extent to which sperm versus seminal plasma mediate beneficial effects on fertility in women.

These observations shed new light on the significance of sperm in the events of conception and female immune adaptation for pregnancy. There is considerable potential for sperm-associated signals to vary in response to genetic and environmental factors, and so impact the quality and strength of the female tract response[72,73]. Our findings raise the prospect that sperm-

**Table 2 Summary of key uterine cytokines regulated by seminal fluid components delivered by intact and vasectomized males in vivo, and sperm and seminal vesicle fluid (SVF) in vitro, and evidence of additive or synergistic interactions between sperm and SVF. Arrow = up- or down-regulated; dash = unchanged.**

| cytokine | in vivo | | in vitro | | | sperm-SVF interaction | |
|---|---|---|---|---|---|---|---|
| | Intact mating | vasectomized mating | sperm | SVF | sperm + SVF | additive | synergistic |
| CCL2 | ↑↑↑ | ↑↑ | ↓ | ↑ | ↑↑↑↑ | | ✓ |
| CSF2 | - | ↑↑ | ↓ | ↑ | | | |
| CSF3 | ↑↑ | - | ↑ | ↑ | ↑↑ | ✓ | |
| CXCL1 | ↑↑↑ | ↑↑ | - | ↑ | | | |
| CXCL2 | ↑↑ | ↑↑ | ↑ | ↑ | ↑↑↑↑ | | ✓ |
| IL6 | ↑↑↑ | ↑ | ↑ | ↑ | ↑↑ | ✓ | |
| LIF | ↑↑ | ↑↑ | ↓ | ↑ | | | |
| TNF | ↑↑ | ↑↑ | ↓ | ↑ | | | |

associated signals contribute to immune-mediated cryptic female choice[2,74], by which females detect and respond to seminal fluid signals during the post-copulatory phase, to selectively increase or decrease the likelihood of a specific male siring pregnancy[10,55]. Sperm have been viewed as a target for hostile cryptic female choice mechanisms, and are known to exhibit within- and between-male variation in features that promote their chance of transiting the tract and fertilizing oocytes[55], but until now, have not been shown to affect female tract receptivity. Recognition that sperm influence reproductive events beyond simply fertilizing oocytes highlights that reduced sperm quality could adversely impact not only fertility, but also implantation and fetal development, and so potentially offers a new target for interventions to manipulate fertility and reproductive outcomes.

## Methods
The experimental approach is summarized in a graphical abstract (Supplementary Fig. 9).

**Mice.** Pathogen-free CBA x C57Bl/6 F1 (CBAF1) and BALB/c female mice, and BALB/c and C57Bl/6 male mice were obtained from Laboratory Animal Services, University of Adelaide (Adelaide, Australia) or Animal Resource Centre (Perth, Australia). Mice with a null mutation in the $Tlr4$ gene ($Tlr4^{-/-}$ mice, on a BALB/c background) were sourced from Prof. Akira (Osaka University, Osaka, Japan)[75] and supplied by Prof. Paul Foster (University of Newcastle, Newcastle, Australia). All mice were co-housed in specific pathogen free conditions at the University of Adelaide Medical School Animal House on a 12 h light–dark cycle and were given standard rodent chow (Envigo, Teklad Diets, Madison WI) and water $ad$ $libitum$. Experimental females were 9–16 wks and males were 12–28 wks of age. Vasectomy was performed at 6–8 wks in BALB/c males under anesthesia with 15 μl/g of Avertin (2,2,2 tribromomethanol, Sigma-Aldrich) in 2-methyl-2-butanol, by ligation of vas deferens then bisection using quarterization. Animal usage was in accordance with the Australian Code of Practice for the Care and Use of Experimental Animals, and experiments were approved by the University of Adelaide Animal Ethics Committee (approval numbers: M-2010-095, M-2014-023, M-2017-006).

**In vivo mating experiments.** For mating experiments, the estrous cycle stage was analyzed by wet mounts of vaginal lavage cells[76]. Females in the pro-estrus stage of the estrus cycle were caged with BALB/c males (1:1 ratio) at 2330–2345 h and the precise time of mating was monitored by video recording as described[13]. At 8 h post mating, or at 2000 h on the day of pro-estrus for unmated controls, females were humanely euthanized and uterine tissue was harvested. For microarray and qPCR experiments, uterine tissue was placed in ice-cold RNase-free PBS, trimmed of fat, mesentery, and blood vessels, then slit lengthwise to expose the endometrial surface. The endometrial layer was scraped into Qiazol RNA lysis solution and homogenized for 2 cycles at 5500 rpm for 30 s (Precellys 24, Bertin Technologies, Saint-Quentin-en-Yvelines Cedex, France). Homogenized samples were stored at −80 °C until further use. For cytokine quantification, endometrial tissues were homogenized in 500 μl PBS containing 1% bovine serum albumin (PBS-BSA) and protease inhibitor cocktail (cOmplete™ Mini, EDTA-free, Sigma-Aldrich, St Louis, Mo, USA). Uterine luminal fluid was flushed with 50 μl PBS-BSA. Insoluble material was pelleted at 14,000 × $g$ for 10 min and the supernatant stored at −80 °C until analysis. For uterine luminal fluid immunohistological staining, fluid was extruded neat from the uterus, and cell smears made on glass slides.

**Isolation of total RNA.** Total RNA was extracted using the miRNeasy extraction kit (Qiagen, Hilden, Germany) and DNase treated using DNA-$free$™ (Thermo Fisher Scientific) as described[13]. RNA was quantified using the nano-drop spectrophotometer and integrity analyses was performed using the Agilent Bioanalyzer (Agilent Technologies, Santa Clara CA). RNA with an RIN > 7 was used for further analysis. RNA was stored at −80 °C until further use.

**Microarray analysis and bioinformatics analysis.** For microarray, endometrial RNA from $n = 16$ individual females was pooled into four independent biological replicates per treatment group ($n = 4$ endometrial samples per replicate). Microarray analysis was performed using Affymetrix Mouse Gene 1.0 ST Arrays at the Adelaide Microarray Centre (Affymetrix, Santa Clara CA). Total RNA (300 ng) was labeled using the Ambion WT Expression Kit as per the manufacturer's instructions (Ambion Inc, Austin TX). The microarray data were analyzed using Partek Genomics Suite (Partek Inc, Chesterfield MO). Briefly, cell files were imported using RMA background correction, Partek's own GC content correction, and mean probe summarization. For microarray data, differential gene expression was assessed by ANOVA with the $P$ value adjusted using step-up multiple test correction to control for the false discovery rate (FDR), as described[77]. All data were included in the analysis. Differentially expressed genes were defined as those with a fold-change ≥ 1.5 and FDR-adjusted $P ≤ 0.05$. Differentially expressed genes were further analyzed by Ingenuity Pathway Analysis (Qiagen) version 47547484. The microarray data in this manuscript is deposited in the National Centre for Biotechnology Information Gene Expression Omnibus and are accessible through GEO series accession number GSE167485.

**Reverse transcription and quantitative PCR.** Total cellular RNA was reverse transcribed from 150 ng random hexamer-primed RNA employing a Superscript-III Reverse Transcriptase kit following the manufacturers' instructions. Primer pairs specific (Supplementary Table 3) for published cytokine sequences were designed using Primer Express Version 2 (Thermo Fisher Scientific). PCR assay optimization and validation experiments were performed using control murine endometrial cDNA. Sequence specificity was confirmed using Sanger sequencing at Australian Genome Research Facility (AGRF, Adelaide node, Australia) of PCR primer products purified from 2% agarose gels. Amplification efficiency was performed using serial dilutions of primers in murine endometrial cDNA. All primer pairs had a correlation coefficient of >0.95 and efficiency of 90–110%.

Quantitative PCR (qPCR) was performed on 20 ng of cDNA, supplemented with optimized PCR primers (Supplementary Table 3) and 1× Power SYBR Green PCR master mix (Thermo Fisher Scientific). The negative control included in each reaction contained $H_2O$ substituted for cDNA or RNA without reverse transcription. PCR amplification was performed in an ABI Prism 7000 Sequence Detection System (Applied Biosystems) using the following conditions: 95 °C for 10 min followed by 40 cycles of 95 °C for 15 s and 60 °C for 1 min. Gene expression was determined[78] using the delta C(t) method using $Actb$ as the reference gene, after confirmation that $Actb$ was unchanged between treatment groups.

**Cytokine enzyme-linked immunosorbent assay and multiplex microbead analysis.** Cytokines secreted into the luminal fluid or from uterine epithelial cell culture supernatants were quantified using either enzyme-linked immunosorbent assay (ELISA) (CXCL7 Duo-set ELISA, R&D systems, Minneapolis, MN) or Merck Milliplex MAP mouse cytokine/chemokine multiplex microbead kit (Merck Millipore, Burlington, MA) following the manufacturer's instructions. Samples for the CXCL7 ELISA were diluted 1:100 in assay buffer and the minimum detectable threshold was 15.6 pg/ml and 7.8 pg/ml, respectively. Duoset ELISA data were analyzed using Bio-Rad Microplate Reader Benchmark and Microplate Manager 5.2.1 Build 106 (Bio-Rad Laboratories). Samples for multiplex were diluted 1:5, 1:10, or 1:100 in assay buffer and the minimum detectable threshold for all analytes was <3.2 pg/ml. Cytokines measured by multibead assay were: CCL2, CSF1, CSF2,

CSF3, CXCL1, CXCL2, CXCL5, CXCL10, IL1B, IL6, LIF, and TNF. Microbead data were acquired on a Luminex 200 instrument and analyzed using eXponent 3.1 (Luminex Corp, Austin, TX).

**t-distributed stochastic neighbor embedding (tSNE) analysis**. For tSNE analysis of uterine mRNA and luminal fluid protein levels, relative mRNA levels of CCL22, CSF2, CSF3, CXCL1, CXCL2, CXCL5, CXCL7, CXCL10, IL1b, IL6, LIF, and TNF from each virgin proestrus control, vasectomized mated or intact mated female was compiled into a matrix as a.csv file. The matrix was opened in FCS Express v6.06 (De Novo Software, California, USA) to carry out the dimensional reduction. Data from each mouse was transformed using 500 iterations of the tSNE algorithm utilizing each gene listed above with a perplexity of 16 for qPCR data and 8 for Luminex data, and a Barns-Hut approximation value of 0.5. The acquired tSNE-Y and tSNE-X values were applied to a dot plot for visualization.

**Immunohistology**. Ly6G+ neutrophils were detected in paraffin-embedded uterine tissue of estrus control, vasectomized-mated or intact-mated females using rat anti-mouse Ly6G RB6-8C5 hybridoma supernatant[29]. Uterine tissues were fixed with 4% paraformaldehyde (wt/vol) in PBS overnight at 4 °C, washed in 1× PBS, and stored in ethanol before paraffin embedding. Tissue sections (5 µM) were cut on a HM 325 Rotary Microtome (Thermo Fisher Scientific), dewaxed in xylene, and rehydrated. Endogenous peroxidase activity was blocked by incubating sections in quenching solution (50% methanol, 10% $H_2O_2$) for 15 minutes at room temperature followed by washing in MilliQ $H_2O$. Non-specific antibody binding was blocked by incubation in blocking buffer (15% normal rabbit serum (vol/vol) in PBS) for 30 min at 37 °C. Sections were washed in PBS before incubation with rat anti-Ly6G, or irrelevant isotype-matched control mAb (1:500 dilution of hybridoma supernatant in 1.5% normal rabbit serum (vol/vol) in PBS, overnight at 4 °C in a humidified chamber. Following washing in PBS, biotinylated rabbit anti-rat IgG (1:500 dilution in antibody dilution buffer; Vector Laboratories, Burlingame, CA) was added for 40 minutes at room temperature. Sections were washed in PBS before incubation with ABC Vectorstain Elite kit (Vector Laboratories) according to manufacturer's instructions. Detection was performed using diaminobenzidine tetrachloride (Sigma-Aldrich), and sections were counterstained with hematoxylin (Sigma-Aldrich). Images were captured using a NanoZoomer 1.0 U (Hamamatsu, Shizuoka, Japan) at a zoom equivalent of a 40× objective lens and quantified using FIJI Image J (Ludwig Institute for Cancer Research, New York, USA).

Complexes containing neutrophils, neutrophil extracellular traps (NETs), and sperm were stained in uterine luminal fluid using DAPI and rat anti-mouse Ly6G RB6-8C5 hybridoma supernatant[30]. Luminal fluid was extruded from the uterus at 8 h post-mating, smeared, and air-dried onto glass microscope slides, then fixed in 96% ethanol at 4 °C for 10 min. Slides were rehydrated and stained to detect Ly6G as above, using rabbit anti-FITC (BD Biosciences, San Jose, CA, diluted 1:200 in antibody dilution buffer), or with 4′,6-diamidino-2-phenylindole, dihydrochloride (DAPI, 300 nM in PBS, Thermo Fisher Scientific).

**Flow cytometry**. Uterine draining para-aortic lymph nodes (PALN) were collected from CBAF1 females in estrus, or on day 3.5 pc following mating with BALB/c males[17]. Single-cell suspensions of PALN were prepared by gentle crushing between two frosted glass slides and cells were washed in complete RPMI media (Thermo Fisher Scientific; RPMI + 10% FCS + 2% penicillin/streptomycin). Cells ($1 \times 10^6$) were plated in a 96-well U bottom plate (Corning, New York), washed twice in PBS then incubated with fixable viability stain 620 (1:1000 in PBS, BD Biosciences) at RT in the dark for 20 min before washing in FACS buffer and incubation with Fc receptor block (as per manufacturer's instructions, BD Biosciences). Cells were then stained with fluorophore-conjugated monoclonal antibodies CD4•APC-Cy7 (GK1.5, 2 µg/mL; BD Biosciences), CD25•PE-Cy7 (PC61, 2 µg/mL; BD Biosciences), and NRP1•BV421 (3E12, 0.3 µg/mL; BioLegend, San Diego, CA) in FACS buffer for 25 min at 4 °C. Cells were fixed and permeabilized using the FOXP3 Staining Buffer Set; Thermo Fisher Scientific) according to manufacturer's instructions. Fixed cells were labeled with FOXP3•APC (FJK-16s, 2 µg/mL; Thermo Fisher Scientific), Ki67•FITC (SolA15, 4 µg/mL; Thermo Fisher Scientific) and CTLA4•PE (UC10-4F10-11, 2 µg/mL; BD Biosciences) diluted in permeabilization wash (FOXP3 Staining Buffer Set; Thermo Fisher Scientific). Data was acquired using a FACS Canto II flow cytometer (BD Biosciences) and analyzed using FlowJo software (Treestar, Ashland, OR, USA). Numbers of cells in subpopulations were calculated as proportions of total live cells.

Doublet discrimination was performed based on the forward and side scatter profiles of the cells, and live/dead gating was performed using a viability dye, to enable analysis of single, live cells respectively (Supplementary Fig. 5). Live lymphocytes positive for CD4 were gated, and Treg cells were defined as CD4+FOXP3+ cells. Gates were established within the CD4+FOXP3+ Treg cell population to distinguish NRP1−/lo (pTreg) and NRP1+ (tTreg) cells. Ki67, CD25, and CTLA4 were measured within total Treg, tTreg, and pTreg populations. In all samples, gates were established using unlabeled and fluorescence minus one controls.

**Sperm and seminal vesicle fluid preparation**. Sperm and seminal vesicle fluid were collected from individually housed BALB/c male mice. Cauda epididymis were excised and placed in 1 ml DMEM + FCS (DMEM, plus 5.5 mM D-glucose, 25 mM HEPES, 0.04 mM phenol red, 1 mM sodium pyruvate, 10% fetal calf serum, 1× antibiotic/antimycotic) (Thermo Fisher Scientific, Waltham MA) at 37 °C, then lacerated and incubated for 10 min to release sperm. Sperm were counted using a hemocytometer and further diluted in DMEM + FCS to $30 \times 10^6$/ml, before immediate addition to uterine epithelial cell cultures. Seminal vesicles were excised and ~50 µl fluid per vesicle was extruded into 450 µl DMEM + FCS (10% seminal vesicle fluid) before immediate addition to uterine epithelial cell cultures.

**In vitro uterine epithelial cell culture**. Uteri were harvested from CBAF1, $Tlr4^{+/+}$, and $Tlr4^{−/−}$ female mice identified as estrous by vaginal lavage cytology, and epithelial cell monolayers (> 90% epithelial cells) were prepared as previously described[13,79]. Enriched epithelial cells were resuspended at $7.5 \times 10^5$ cells/ml in DMEM + FCS, and 150 µl aliquots were plated in 48-well multidishes (Nunc, Roskilde, Denmark). Cells were incubated for 4 h at 37 °C in 5% $CO_2$ to permit adherence, then sperm ($3.8 \times 10^5$, $7.5 \times 10^5$, or $1.5 \times 10^6$ /well to give 1:2, 1:1, and 2:1 sperm to uterine epithelial cell ratio), seminal vesicle fluid (1:2 dilution of 10% seminal vesicle fluid in DMEM + FCS), or a combination of both sperm and seminal vesicle fluid (5% seminal vesicle fluid and $1.5 \times 10^6$ sperm/well) were added to a final volume of 200 µl. Supernatants were collected after 16 h incubation, and following replacement of culture medium and a further 24 h incubation, then centrifuged at $400 \times g$ to remove cellular debris, and stored at −80 °C until cytokine assay. Adherent cells were quantified by Rose Bengal dye uptake (0.25% in PBS, 10 min at room temperature) (Sigma-Aldrich) and cell lysis in 0.1% SDS by measuring absorbance at 570 nm using a BioTek Synergy H1 Hybrid Multi-Mode Reader and BioTek Gen5 software (BioTek Instruments, Inc., Winooski, VT, USA) as described previously[13,79].

**Statistical analysis**. qPCR, cytokine protein, flow cytometry, and immunohistochemical data were analyzed using GraphPad Prism, version 9.0.0 software (GraphPad Software LLC, San Diego CA). Outlier values were identified using the ROUT method (Q < 1%) and removed from analyses. Data were first assessed for normality of distribution using the D'Agostino and Pearson normality test. Normally distributed data were analyzed by one-way ANOVA with a post-hoc two-sided Sidak t-test to detect differences between treatment groups. Data not normally distributed were transformed by Box-Cox transformation[80] and analyzed as above. If transformation did not yield normal distribution, data were analyzed by non-parametric Kruskal Wallis test with post-hoc Dunn's two-sided multiple comparisons test. In vitro cell culture cytokine data was analyzed using SPSS for Windows, version 20.0 software (SPSS Inc, Chicago, IL). All data were included in the analysis. Differences between groups were considered significant when $P < 0.05$.

**Reporting summary**. Further information on research design is available in the Nature Research Reporting Summary linked to this article.

## Data availability
The microarray data in this manuscript is deposited in the National Centre for Biotechnology Information Gene Expression Omnibus and are accessible through GEO series accession number GSE167485. The authors declare that all other data supporting the findings of this study are available within the paper and its supplementary information files.

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

## Acknowledgements

This study was supported by Project Grant APP1041335 (to S.A.R.) from the National Health and Medical Research Council (Australia) and Discovery Grants DP0986882 and DP160102366 (to S.A.R.) from the Australian Research Council.

## Author contributions

S.A.R. and J.E.S. designed research; J.E.S., D.J.S, E.S.G., H.Y.C., R.A.M., and L.M.M. performed research; J.E.S., D.J.S, E.S.G., H.Y.C., R.A.M., and L.M.M. analyzed data; and S.A.R and J.E.S. wrote the paper.

## Competing interests

The authors declare no competing interests.
