## [Peer Review File · Communications Biology]

Reviewers' comments:

Reviewer #1 (Remarks to the Author):

The authors previously showed that mouse seminal plasma affects immune gene expression in the mouse female reproductive tract. Here, they test whether sperm also cause effects on female gene expression. They compare uterine gene expression among unmated females, females mated to intact males, and females mated to vasectomized males. They find that some genes' expression differs between mates of intact vs. vasectomized males. They then test in vitro whether these genes respond to epididymal sperm in uterine epithelial cells. They interpret their results as indicating that sperm affect immunity in mated female mice. Their results suggest that TLR4 signaling plays a role in this process, and using mutants they demonstrate this.

The RNA work, by microarray, appears to have been done well, with qPCR and/or protein-level verification. The results are believable. The Discussion touches on a variety of topics of relevance, such as exosomes. The paper is well written.

Some items need addressing:

1. The authors published a paper in 2011 that also showed differences in uterine gene expression between females that mated to intact vs vasectomized males. Please explain how the present paper goes beyond the earlier one, and how they overlap or confirm.
2. Intact males are not surgically-treated. This difference between them and vasectomized males is not considered in the interpretations. Might there be behavioral or other differences that result in vasectomized males transferring a different amount of soluble seminal plasma as compared to intact males?
3. Vasectomized males lack sperm but also secretions from testes and epididymis. So it is too strong to say that sperm, alone, causes the gene expression changes. Sometimes (line 171) the authors acknowledge this but at other places (line 30, 277) they don't.
4. In the in vitro experiment the sperm are from the epididymis, so not all of them are fully mature. In addition, epididymal secretions or exosomes are likely present in those sperm samples. The experiments and data are fine, but I suggest the authors moderate the interpretation to include brief consideration of these points.

Minor:

1. Do any of the genes show evidence of opposing effects of soluble factors and sperm?
2. In line 244, "communicating" implies an active role of sperm, but this is not shown. Please reword.

Reviewer #2 (Remarks to the Author):

Commendable work, following a plethora of previous studies and similar studies in other species. The paper reads well and it is consistent. A single point appears: vasectomy implies that not only spermatozoa are deleted from the experimental layout, but also the caudal epididymis fluid, whose role/s have been ascribed to be directly related to sperm function but also to strongly interact with the female genital lining, since it is the first fluid that encounters the female genital internal tract. This reviewer considers that the paper ought to contain a stronger consideration of this aspect and the implications involved. I foresee such integration to be present in a revised version.

Reviewer #3 (Remarks to the Author):

The manuscript entitled "Sperm modulate immune mechanisms of female reproductive investment" by

Schjenken et al., provides an interesting new information of the endometrial immune responses to sperm in mouse. The in vivo and in vitro investigations showed that sperm provoked a pro-inflammatory cytokine synthesis in the uterine epithelium probably via TLR4 signaling pathway. Moreover, sperm induced neutrophils recruitment into the endometrial stromal tissue in order to remove excess and dead sperm and cell debris. In addition, sperm triggered activation and expansion of Treg cells in lymph nodes draining the uterus which are involved in the subsequent endometrial receptivity and embryo tolerance.

Together, the manuscript is well-written and provides a concept for the impact of sperm, in addition to seminal plasma, as paternal antigens in modulating the uterine immune microenvironment for embryo receptivity and tolerance. However, several major concerns need to be addressed regarding the experimental design, results and discussion sections.

Major concerns:

1. To identify the definite impact of sperm on uterine immune responses, the authors investigated the gene expression and cytokines in the mouse uterus after mating with intact or vasectomized males. Comparing the results after mating with intact and vasectomized males, the authors declared that the differences in the results might be due to sperm effects and this effect was mediated by TLR4 pathway. Logically, we cannot exclude the possibility of interactions between sperm and seminal fluids in intact male, since the in vitro system showed clear additive and synergistic effects between sperm and seminal vesicle fluid. Additionally, ejaculated sperm are coated by several major proteins originated from seminal plasma compared to epididymal sperm. Therefore, to investigate in vivo sperm effect alone, I strongly recommend to give an insemination by epididymal sperm then investigate the differential changes in gene expression and cytokines in the mouse uterus.
2. The data do not strongly support the hypothesis that TLR4 signaling pathway acts as a main signaling pathway for sperm-induced inflammatory responses in uterine epithelial cells. Specifically the authors previously reported that seminal plasma interacts with uterine epithelium through TLR4 and their components might bind to sperm surface after ejaculation. To confirm this hypothesis, further investigations are needed to identify major TLR4 signaling proteins and endogenous sperm ligands involved as well as the impact of activation or blockage of TLR4 pathway on sperm-induced inflammation in endometrial epithelium.
3. The authors collected the uterus 8 h after mating and investigated gene and protein expression. On which basis, this time point was selected. It is well established in most animal species and human, that insemination triggers transit and rapid inflammatory responses in endometrial tissue within 2-4 h. Did the authors make a preliminary time-dependent experiment to investigate the uterine immune response after mating? If so, please show the data, or explain some detail from own previous published paper. Similarly, the authors selected 16 and 40 h for investigating sperm and seminal vesicle fluid effects on uterine epithelial cells in vitro. On which basis these time points were selected? Moreover, coculture of uterine epithelial cells with sperm and relatively high concentration of seminal vesicle fluid (5%) for 40 h is extremely long incubation time. Did the author checked cell viability after the end of the co-culture??
4. The authors collected uterine fluid 8 h after mating and measured cytokines concentrations. Why the authors did not investigated neutrophils influx into the uterine luminal fluid. It might provide very interesting information more than their distribution in the uterine stoma.
5. The authors added sperm to endometrial epithelial cells from Tlr4^{-/-} mice to investigate the role of TLR4 signaling pathway in sperm-triggered inflammation in uterus in vitro. However, it might be more significant to test this hypothesis in vivo using Tlr4^{-/-} mice model.

Minor concerns

Abstract

1. Line 23 – This word 'attenuate' is not suitable here. Because in the previous sentence it is mentioned as modulate. So, the 'effect' probably suitable here.
2. Line 24 – which gene expression?
3. Line 29 – Change 'In vitro' in to italic

Introduction

1. Line 40 – Replace the word 'conditions' with 'failures'
2. Line 85 – which gene expression?

Results

1. Labelling of Fig. 4 is too small thus hard to read. The figure 4 has to be created larger including the labeling of the axis.
2. Line 98 - which gene expression? Please specify.
3. Line 101 – no need 'may also'
4. In the in vivo mice study, gene expression and protein analysis were conducted at 8 h following mating. The significance of this time point should be mentioned in the discussion section.
5. At 8 h following mating did sperm present in the uterus? If so, did authors make sure to remove sperm from epithelial cells before doing RNA extraction? If not, did authors clarify whether sperm responsible for any gene expression?
6. In the in vitro study, it is not clearly mentioned the concentration and sperm preparation method.
7. Line 110/122 – it is mentioned here that 697 genes were upregulated (also in Fig. 1B), but in line 122 it is mentioned as 698 genes. Which one is correct?
8. Line 154 – Please specify which role. Maybe 'immune regulating role'?
9. Line 227: (Fig.7A) ?? Probably you mean (Fig. 5A)?
10. In in vitro uterine epithelial cell culture model, sperm were co-cultured with epithelial cells in DMEM medium. Did authors confirm whether sperm are active in these conditions?
11. It is not easy to follow up the results for gene expression and cytokines concentrations in uterine luminal fluid and uterine tissues from Fig. 2 and supplementary Figs. 1, 2, and 3. I suggest that this data should be presented together as main data.
12. In Fig. 4 and supplementary Fig. 7, it might be helpful for the readers to know the net concentrations of such cytokines for control group and accordingly other groups (% control). I strongly

recommend to describe this information in figure legends.

13. Line 257-259: As long as I know, this article (#31) did not use a TLR4 antagonist, but antibody. Also, those authors emphasized high linkage of TLR2-mediated inflammation by sperm, rather than TLR4. Maybe species difference there.

14. The discussion section seems to me too long. It should be shortened, precisely interpret available results, and avoid too many speculations.

We thank the Editor and Reviewers for their thoughtful and constructive comments. We have addressed these in a point-by-point response below.

Reviewer #1 (Remarks to the Author):

The authors previously showed that mouse seminal plasma affects immune gene expression in the mouse female reproductive tract. Here, they test whether sperm also cause effects on female gene expression. They compare uterine gene expression among unmated females, females mated to intact males, and females mated to vasectomized males. They find that some genes' expression differs between mates of intact vs. vasectomized males. They then test in vitro whether these genes respond to epididymal sperm in uterine epithelial cells. They interpret their results as indicating that sperm affect immunity in mated female mice. Their results suggest that TLR4 signaling plays a role in this process, and using mutants they demonstrate this.

The RNA work, by microarray, appears to have been done well, with qPCR and/or protein-level verification. The results are believable. The Discussion touches on a variety of topics of relevance, such as exosomes. The paper is well written.

Some items need addressing:

1. The authors published a paper in 2011 that also showed differences in uterine gene expression between females that mated to intact vs vasectomized males. Please explain how the present paper goes beyond the earlier one, and how they overlap or confirm.

The author is likely referring to our publication "Guerin LR, Moldenhauer LM, Prins JR, Bromfield JJ, Hayball JD, Robertson SA. Seminal fluid regulates accumulation of FOXP3+ regulatory T cells in the preimplantation mouse uterus through expanding the FOXP3+ cell pool and CCL19-mediated recruitment. *Biol Reprod* **85**, 397-408 (2011)." We have added a comment to indicate that this earlier study evaluated only a very limited number (n=8) of immune regulatory genes involved in control of regulatory T cells (Tregs), and the time point was day 3.5 pc, not day 0.5 pc as in the current study (line 77-81): "We have previously reported that the Treg cell-specific transcription factor *Foxp3*, as well as a critical Treg cell-attracting chemokine *Ccl19*, are more strongly expressed at the time of embryo implantation in the uterus of mice earlier exposed to seminal fluid of intact males, as opposed to vasectomized males¹⁸, implying that stronger immune tolerance may be generated when sperm are present." The same genes would not be expected to appear in the differentially expressed genes in the current study, as newly recruited Tregs do not appear in the uterus until just prior to embryo implantation.

The reviewer may also be referring to our earlier publication "Schjenken JE, Glynn DJ, Sharkey DJ, Robertson SA. TLR4 signaling is a major mediator of the female tract response to seminal fluid in mice. *Biol Reprod* **93**, 68 (2015)" where we used microarray and qPCR to investigate the effect of whole seminal fluid on uterine gene expression in mated mice. The previous paper (Schjenken et al. 2015) which shows uterine genes induced by whole seminal fluid, but did not investigate the significance of sperm. We have added sentences to the discussion (lines 280-81, and line 285) to reinforce how the current study is a significant advance.

2. Intact males are not surgically-treated. This difference between them and vasectomized males is not considered in the interpretations. Might there be behavioral or other differences that result in

vasectomized males transferring a different amount of soluble seminal plasma as compared to intact males?

We have added a sentence to the discussion (line 290-291) to acknowledge this issue, which we accept is a valid limitation of the study. However, that *in vitro* experiments demonstrate sperm-mediated induction of cytokines from uterine epithelial cells, confirms the conclusion that sperm account in large part for the difference in female response elicited by vasectomised versus intact males. We and others have utilised vasectomised males over many decades and have no evidence of any behavioural difference (in regard to mating behaviour or plug formation), and embryo transfer experiments show females mated with vasectomised males are receptive to embryo implantation.

3. Vasectomized males lack sperm but also secretions from testes and epididymis. So it is too strong to say that sperm, alone, causes the gene expression changes. Sometimes (line 171) the authors acknowledge this but at other places (line 30, 277) they don't.

We acknowledge and clearly state that the *in vivo* gene expression experiment points to a role for sperm, but is not conclusive, as testes and epididymis secretions might also contribute (now line 153, lines 204, 288-290). When inferences are drawn from the *in vivo* experiment, the interpretation is qualified to include potential contributions of epididymal secretions and sperm. We have added a very specific statement on this at line 288 "Whether epididymal secretions interact with female tissues *in vivo* is not clear ³⁶, but is likely given that microvesicles in seminal fluid can modulate gene expression in female reproductive tract cells *in vitro* ^{37,38}."

In contrast, we contend that the data from *in vitro* experiments, where we incubate epididymal sperm with uterine epithelial cells, provide conclusive evidence that sperm are capable of and contribute to uterine cytokine induction. When inferences are drawn from the *in vitro* experiment, or from the experiments collectively, we argue that the conclusion that sperm are specifically implicated in influencing uterine gene expression, is warranted. We have taken great care with the wording of all sentences to reflect these interpretations and not draw unsupported conclusions. We have carefully revised the wording at all relevant sentences in the manuscript to qualify the conclusions appropriately.

Line 30 is amended to read "Collectively these experiments show that sperm assist in promoting female immune tolerance by eliciting uterine cytokine expression through TLR4-dependent signaling." Line 277 (now lines 292-293) is revised to read: "However, our *in vitro* experiment clearly demonstrates that sperm can bind and transmit signals to uterine epithelial cells, to elicit increased cytokine synthesis." And (line 295-298): "Collectively therefore, these data provide compelling evidence of a sperm-mediated effect, and imply that the constrained response to vasectomized males is at least partly attributable to sperm."

4. In the in vitro experiment the sperm are from the epididymis, so not all of them are fully mature. In addition, epididymal secretions or exosomes are likely present in those sperm samples. The experiments and data are fine, but I suggest the authors moderate the interpretation to include brief consideration of these points.

We agree with the reviewer on the point regarding the maturation status of sperm, and the possibility that epididymal sperm are less mature and potentially have different surface structures that could alter their capacity to elicit cytokines from female uterine epithelial cells. We had noted this in lines 310-312, and now add a sentence at line 313-315: "This raises a limitation of the *in vitro* experiments, which used epididymal sperm that could have minor differences in surface properties to ejaculated sperm."

The sperm used in the in vitro experiments were prepared using a swim-out protocol, so any exosomes would be so extensively diluted (~1:2000) as to have negligible effect in the in vitro experiment. We have added a sentence at line 293-295: "The protocol for sperm recovery caused extensive (~2000-fold) dilution of any epididymal secretions in the *in vitro* experiments described herein."

Minor:

1. *Do any of the genes show evidence of opposing effects of soluble factors and sperm?*

Amongst the 19 cytokine and chemokine genes identified as differentially expressed (>1.4-fold fold change) in endometrial tissue of intact (int) compared to vasectomized (vas) mated females, all were more strongly differentially expressed (18 upregulated, 1 downregulated) after mating with intact versus vasectomised males. A similar pattern was confirmed by qPCR for 13 of these 19 genes. Similarly, in the in vitro experiments, none of the cytokines measured showed opposing effects of seminal plasma and sperm. We have added a sentence at line 319 to state: "We did not find uterine cytokines where sperm and seminal plasma exerted opposing effects, as was suggested by a previous study in pigs".

2. *In line 244, "communicating" implies an active role of sperm, but this is not shown. Please reword.*

'communicating' is changed to 'interacting'. Thankyou for this recommendation.

Reviewer #2 (Remarks to the Author):

Commendable work, following a plethora of previous studies and similar studies in other species. The paper reads well and it is consistent. A single point appears: vasectomy implies that not only spermatozoa are deleted from the experimental layout, but also the caudal epididymis fluid, whose role/s have been ascribed to be directly related to sperm function but also to strongly interact with the female genital lining, since it is the first fluid that encounters the female genital internal tract. This reviewer considers that the paper ought to contain a stronger consideration of this aspect and the implications involved. I foresee such integration to be present in a revised version.

We thank the reviewer for this important point. We have addressed this in response to Reviewer 1, point #3. In particular, we have added a sentence on this issue to the discussion, lines 288-290: "Whether epididymal secretions interact with female tissues is not clear³⁵, but is likely given that microvesicles in seminal fluid can modulate gene expression in female reproductive tract cells^{36,37}."

Reviewer #3 (Remarks to the Author):

The manuscript entitled "Sperm modulate immune mechanisms of female reproductive investment" by Schjenken et al., provides an interesting new information of the endometrial immune responses to sperm in mouse. The in vivo and in vitro investigations showed that sperm provoked a pro-inflammatory cytokine synthesis in the uterine epithelium probably via TLR4 signaling pathway. Moreover, sperm induced neutrophils recruitment into the endometrial stromal tissue in order to remove excess and dead sperm and cell debris. In addition, sperm triggered activation and expansion of Treg cells in lymph nodes draining the uterus which are involved in the subsequent endometrial receptivity and embryo tolerance.

Together, the manuscript is well-written and provides a concept for the impact of sperm, in addition to seminal plasma, as paternal antigens in modulating the uterine immune microenvironment for embryo receptivity and tolerance. However, several major concerns need to be addressed regarding the experimental design, results and discussion sections.

Major concerns:

1. To identify the definite impact of sperm on uterine immune responses, the authors investigated the gene expression and cytokines in the mouse uterus after mating with intact or vasectomized males. Comparing the results after mating with intact and vasectomized males, the authors declared that the differences in the results might be due to sperm effects and this effect was mediated by TLR4 pathway. Logically, we cannot exclude the possibility of interactions between sperm and seminal fluids in intact male, since the in vitro system showed clear additive and synergistic effects between sperm and seminal vesicle fluid. Additionally, ejaculated sperm are coated by several major proteins originated from seminal plasma compared to epididymal sperm. Therefore, to investigate in vivo sperm effect alone, I strongly recommend to give an insemination by epididymal sperm then investigate the differential changes in gene expression and cytokines in the mouse uterus.

We acknowledge the potential contribution of epididymal secretions and their likely interaction with sperm, and have addressed this in response to Reviewer 1, point #3, and added relevant comments to the manuscript text. We thank the reviewer for the suggestion of this interesting experiment, which we have carefully considered and attempted. On balance, we have decided not to pursue this approach for the following reasons:

1. There are technical challenges of artificial insemination in mice that we cannot readily circumvent. Importantly, we have discovered that the physical manipulation of the female mice that is required to conduct artificial insemination in vivo, alters the experimental endpoints. We find that introduction of a catheter across the cervix, with or without intromission of carrier PBS or other agents, is sufficient to substantially upregulate uterine expression of IL6 and CXCL2, two of the key cytokines that are induced by sperm.
2. The sperm that would be required for this experiment would need to be recovered from the cauda epididymis, and as discussed in the response to Reviewer #1, there is likely a difference in the surface structures of epididymal sperm compared to ejaculated sperm.
3. We are not convinced that the in vivo experiment would achieve data that is different to the results of the in vitro experiment – and given the above considerations, the in vitro approach has advantages.

2. The data do not strongly support the hypothesis that TLR4 signaling pathway acts as a main signaling pathway for sperm-induced inflammatory responses in uterine epithelial cells. Specifically the authors previously reported that seminal plasma interacts with uterine epithelium through TLR4 and their components might bind to sperm surface after ejaculation. To confirm this hypothesis, further investigations are needed to identify majors TLR4 signaling proteins and endogenous sperm ligands involved as well as the impact of activation or blockage of TLR4 pathway on sperm-induced inflammation in endometrial epithelium.

We do not contend that TLR4 is the only signalling pathway utilised by sperm in interacting with female reproductive tract cells, only that it is at least one important mediator. We do not see any alternative logical interpretation of the data we have shown, other than that TLR4 has a key role. The reviewer raises interesting questions about the identity of the TLR4 ligands associated with

sperm, and the nature of the signalling mechanism that is activated. We are pursuing these questions, but have discovered that the biology is rather complex and will require multiple complementary strategies to unravel. It is well beyond the scope of the current study to demonstrate the identity of the ligands or receptor signalling components, and this will be the subject of future publications.

As for other approaches to prove the requirement for TLR4 in sperm-mediated signalling, we cannot think of a better experiment than the gene expression, bioinformatics, and genetic approaches that we have already deployed. We have previously shown that activation of TLR4 in uterine epithelial cells induces all of the same cytokines that are induced by sperm, in an experiment using the model TLR4 ligand bacterial lipopolysaccharide (LPS), published in Schjenken et al (2015). We have added a statement and reference to that effect (line 307): “The model TLR4 ligand lipopolysaccharide (LPS) can induce cytokine release from uterine epithelial cells *in vitro*, but there is insufficient LPS in seminal fluid to be responsible¹³.” As far as blocking antibody experiments, there is little advantage of using other approaches of blocking or neutralising TLR4, above the current genetic approach we have used.

3A. The authors collected the uterus 8 h after mating and investigated gene and protein expression. On which basis, this time point was selected. It is well established in most animal species and human, that insemination triggers transit and rapid inflammatory responses in endometrial tissue within 2-4 h. Did the authors make a preliminary time-dependent experiment to investigate the uterine immune response after mating? If so, please show the data, or explain some detail from own previous published paper.

We have provided a time course experiment to show that 8h is the best time point to evaluate neutrophil changes in the uterine endometrium (Figure 3A). For cost reasons, we were unable to use more than one time point for the microarray experiment. This time point is the same as our previous microarray study and qPCR studies showing that cytokine mRNAs are strongly upregulated in the uterine endometrium at this time (Schjenken et al. Biol Reprod 2015; Bromfield et al PNAS 2014). We have added a comment to state this (line 102-103): “Our previous studies showed strongly upregulated cytokine expression at this time point^{7,13}.” This time point is consistent with studies in many tissues showing that upregulation of inflammatory genes peaks at 4h – 12h after application of an inflammatory trigger, depending on the specific cytokine evaluated. We have conducted time course experiments of seminal fluid effects on human cervical and uterine cells, and have concluded that while different cytokines have different kinetics of induction, when a single time point is required, 8h is the best single time point to use.

3B. Similarly, the authors selected 16 and 40 h for investigating sperm and seminal vesicle fluid effects on uterine epithelial cells in vitro. On which basis these time points were selected? Moreover, coculture of uterine epithelial cells with sperm and relatively high concentration of seminal vesicle fluid (5%) for 40 h is extremely long incubation time. Did the author checked cell viability after the end of the co-culture??

The time points for the *in vitro* coculture experiments were based on extensive experience with similar experiments in our lab over many years. We have used this protocol in previous published studies in reputable journals (eg. Sharkey et al. Biol Reprod, 2018). We have now added a statement to this effect at line 210: “Mouse uterine epithelial cells from estrous mice were cultured with epididymal sperm, seminal vesicle fluid as a comparison, or a combination of both, using an established methodology and time course²⁶”. The timing is different to the *in vivo* gene expression experiment, as protein accumulation takes longer than upregulation of cytokine genes. The sperm

and seminal plasma are incubated only for 16 h, then removed for the latter 24 h of culture. Cytokines induced in response to sperm and/or seminal vesicle fluid induction are measured in supernatants collected both at 16 h and 40 h. We routinely measure cell viability at the end of each experiment and do not see any impact of either treatment on epithelial cell viability. Seminal vesicle fluid-induced cell death can occur if seminal vesicle fluid is collected or processed in a suboptimal manner.

4. The authors collected uterine fluid 8 h after mating and measured cytokines concentrations. Why the authors did not investigate neutrophil influx into the uterine luminal fluid. It might provide very interesting information more than their distribution in the uterine stroma.

We agree with the reviewer that it is useful to quantify neutrophils in the luminal fluid. We have now included an additional experiment on this parameter. Since cellular and soluble material in the luminal cavity is not readily detectable in tissue sections, it was necessary to directly analyse the luminal fluid after extrusion from the cavity. We found that large numbers of neutrophils were readily detectable in smears of uterine fluid from females mated with intact males, but negligible numbers were present after mating with vasectomised males (new Figure panels 3D, E). The vast majority of these neutrophils (detected with antibody to neutrophil marker Ly6G) are degranulated and have formed extensive neutrophil extracellular traps (NETs) amongst which large numbers of sperm are entrapped. Using this approach, it was technically challenging to quantify neutrophils in the luminal fluid with precision. However since there were qualitative differences (large numbers in luminal fluid of females mated with intact males, no detectable neutrophils in luminal fluid of females mated with vasectomised males), we have now reported these observations as additional sentences in the results section (line 182-186): "Large numbers of Ly6G+ neutrophils and extensive complexes of sperm engulfed in Ly6G+ neutrophil extracellular traps (NETs) were also abundant in the luminal cavity as detected by staining with DAPI and Ly6G (Fig. 3D, E). Neither neutrophils nor NETs were present in the luminal fluid of estrus females or females mated with vasectomised males."

5. The authors added sperm to endometrial epithelial cells from Tlr4-/- mice to investigate the role of TLR4 signaling pathway in sperm-triggered inflammation in uterus in vitro. However, it might be more significant to test this hypothesis in vivo using Tlr4-/- mice model.

We appreciate the Reviewer's suggestion to utilise TLR4-/- mice to conduct in vivo experiments. We have provided one new experiment to demonstrate that TLR4 is required for the neutrophil recruitment into the endometrium after mating. This experiment confirms that without TLR4, sperm and seminal plasma-induced cytokines are unable to recruit neutrophils into the reproductive tract after mating. We have carefully considered the utility of this model for other experiments to address the hypothesis that is the subject of the current paper. For the reasons described in the response to point #1 above, there are difficulties that prevent administration of sperm alone into the uterine cavity in vivo. As well, we have utilised the mice to investigate the related hypothesis that TLR4 is important for the female immune response induced by seminal fluid at conception, and the findings are consistent with the experiments described herein. Nevertheless, we do not wish to report those experiments in the current paper, as both seminal plasma and sperm utilise TLR4 to induce cytokine expression in the female reproductive tract, and it is not straightforward to design experiments that distinguish the effects of sperm from seminal plasma separately. Therefore, the additional experiments that we have conducted in vivo in TLR4-/- mice will be reported separately in a forthcoming manuscript.

Abstract

1. Line 23 – This word ‘attenuate’ is not suitable here. Because in the previous sentence it is mentioned as modulate. So, the ‘effect’ probably suitable here.

‘attenuate’ is replaced with ‘affect’

2. Line 24 – which gene expression?

‘global’ is added to read ‘global gene expression’

3. Line 29 – Change ‘In vitro’ in to italic

We have corrected this, thanks.

Introduction

1. Line 40 – Replace the word ‘conditions’ with ‘failures’

‘conditions’ is replaced with ‘disorders’

2. Line 85 – which gene expression?

‘global’ is added to read ‘global gene expression’

Results

1. Labelling of Fig. 4 is too small thus hard to read. The figure 4 has to be created larger including the labeling of the axis.

We have increased the size of Figure 4. The font size of axis labels is the same size as used in all other Figures.

2. Line 98 - which gene expression? Please specify.

‘global’ is added to read ‘global gene expression’

3. Line 101 – no need ‘may also’

‘may also’ is deleted.

4. In the *in vivo* mice study, gene expression and protein analysis were conducted at 8 h following mating. The significance of this time point should be mentioned in the discussion section.

See response to Point 3A above. This time point is the same as our previous microarray study and qPCR studies showing that cytokine mRNAs are strongly upregulated in the uterine endometrium at this time (Schjenken et al. Biol Reprod 2015; Bromfield et al PNAS 2014). We have added a comment to state this (line 102-103): “Our previous studies showed strongly upregulated cytokine expression at this time point ^{7, 13}.”

5. At 8 h following mating did sperm present in the uterus? If so, did authors make sure to remove sperm from epithelial cells before doing RNA extraction? If not, did authors clarify whether sperm responsible for any gene expression?

Contaminating mRNAs carried by sperm are highly unlikely to contribute to the altered gene expression in uterine tissue after mating with intact males. Uterine tissues were extensively washed in PBS before endometrial tissue was recovered. Furthermore it is well known that sperm contain very little mRNA (~100 fg per sperm, Concha et al., 1993 PMID: 8440334). Additionally we included sperm-only controls in in vitro experiments and showed that both cytokine mRNA and protein were below detectable limits in all experiments.

6. In the in vitro study, it is not clearly mentioned the concentration and sperm preparation method.

We have added details on this to the Supplemental Materials and Methods, as follows: "Sperm and seminal vesicle fluid were collected from individually housed BALB/c male mice. Cauda epididymis were excised and placed in 1 ml DMEM+FCS (DMEM, plus 5.5 mM D-glucose, 25 mM HEPES, 0.04 mM phenol red, 1 mM sodium pyruvate, 10% fetal calf serum, 1x antibiotic/antimycotic) (Thermo Fisher Scientific, Waltham MA) at 37°C, then lacerated and incubated for 10 min to release sperm. Sperm were counted using a hemocytometer and further diluted in DMEM+FCS to 30 x 10⁶/ml, before immediate addition to uterine epithelial cell cultures. Seminal vesicles were excised and ~50 µl fluid per vesicle was extruded into 450 µl DMEM+FCS (10% seminal vesicle fluid) before immediate addition to uterine epithelial cell cultures. "

7. Line 110/122 – it is mentioned here that 697 genes were upregulated (also in Fig. 1B), but in line 122 it is mentioned as 698 genes. Which one is correct?

Line 122 is now corrected to read 697 genes.

8. Line 154 – Please specify which role. Maybe 'immune regulating role'?

We have added the word 'immune-regulating' as suggested by the reviewer at line 155.

9. Line 227: (Fig.7A) ?? Probably you mean (Fig. 5A)?

Yes, Fig 5A is correct. Thank you for picking this up.

10. In in vitro uterine epithelial cell culture model, sperm were co-cultured with epithelial cells in DMEM medium. Did authors confirm whether sperm are active in these conditions?

The epididymal sperm co-cultured with epithelial cells in DMEM-FCS remained active and healthy for several hours (>6 h), as indicated by a high motility and capacity to attach and detach from epithelial cells (confirmed by high resolution confocal microscopy).

11. It is not easy to follow up the results for gene expression and cytokines concentrations in uterine luminal fluid and uterine tissues from Fig. 2 and supplementary Figs. 1, 2, and 3. I suggest that this data should be presented together as main data.

We appreciate the reviewer's comment and agree that it would be nice to show all the different n=18 cytokines, at both mRNA and protein level, in the main paper. However realistically, it is not possible to do this without adding another at least 3 large Figures to the main paper, which would mean a total of 10 main Figures. We have carefully considered different options and have decided to stay with the layout for Figure 2 (plus 3 supplemental Figures) provided in the original manuscript. Our reasoning is that this best allows the reader to focus on the key cytokines IL-6, CSF3, and CXCL2 that we later prove (in the in vitro experiments) to be regulated by sperm. If all the cytokines were shown in the main paper, this would dilute the key message around these key cytokines.

12. In Fig. 4 and supplementary Fig. 7, it might be helpful for the readers to know the net concentrations of such cytokines for control group and accordingly other groups (% control). I strongly recommend to describe this information in figure legends.

The average net concentrations of all cytokines in the control (medium only) wells is provided in Figure Legends for Fig. 4 (now Fig. 5), and Supplementary Fig. 7, as suggested by the reviewer. A statement is also added to Figure Legends for Fig. 6 and Supplementary Fig. 8, referring the reader to the previous Figure legend for baseline cytokine concentrations.

13. Line 257-259: As long as I know, this article (#31) did not use a TLR4 antagonist, but antibody. Also, those authors emphasized high linkage of TLR2-mediated inflammation by sperm, rather than TLR4. Maybe species difference there.

We have corrected the statement at line 271, to state that neutralising antibodies to TLR2 and TLR4 were used in the previous study.

14. The discussion section seems to me too long. It should be shortened, precisely interpret available results, and avoid too many speculations.

We substantially shortened the discussion (by ~20%) as recommended by the reviewer. With the addition of new information and comments as requested by both reviewers, the final length of the discussion is reduced by ~10%.

REVIEWERS' COMMENTS:

Reviewer #1 (Remarks to the Author):

The authors responded fully to my comments. The text modifications they made clarify how this work extends their prior findings and acknowledge where technical issues limit interpretation.

Reviewer #3 (Remarks to the Author):

The authors clearly addressed the comments raised by all reviewers in a point-by-point response. Moreover, they added additional experiments to clarify the relative contribution of sperm in modulating uterine immune responses and the role of TLR4 signaling pathway in mediating sperm-induced inflammation in the uterus. The current version of the manuscript is greatly improved and the discussion and conclusion was rephrased based on their findings.

I have just one suggestion. Since the manuscript provides several interesting findings using different biological approaches, it might be interesting for the readers if the authors can draw a diagrammatic illustration for the whole experimental design demonstrating different experimental approaches, both in vivo and in vitro studies, tested samples, used methodology, and the rationale behind each.

We thank the Editor and Reviewers for their thoughtful and constructive comments. We have addressed the remaining issue from Reviewer #3 in a response below.

In addition, we have now also added the GEO accession number to provide access to the microarray data that has been deposited in the National Centre for Biotechnology Information Gene Expression Omnibus. The data is accessible through GEO series accession number GSE167485. This information is stated in the main text at line 410 (Materials and Methods), in the 'Data Availability Statement' at line 441. Additionally it is provided in the Supplementary Information (page 2).

Reviewer #3 (Remarks to the Author):

I have just one suggestion. Since the manuscript provides several interesting findings using different biological approaches, it might be interesting for the readers if the authors can draw a diagrammatic illustration for the whole experimental design demonstrating different experimental approaches, both in vivo and in vitro studies, tested samples, used methodology, and the rationale behind each.

RESPONSE: We have constructed a Graphical Abstract that summarises the experimental approach and summarises the key findings. This is provided as Supplementary Figure 9. The Figure is referred to in the main text at line 390 (Materials and Methods).